# RRNCO: Towards Real-World Routing with Neural Combinatorial Optimization

**Jiwoo Son**[1]*, **Zhikai Zhao**[2]*, **Federico Berto**[2,3]*, **Chuanbo Hua**[2],
**Zhiguang Cao**[4], **Changhyun Kwon**[1,2], **Jinkyoo Park**[1,2]

[1]Omelet   [2]KAIST   [3]Radical Numerics   [4]Singapore Management University

## Abstract

The practical deployment of Neural Combinatorial Optimization (NCO) for Vehicle Routing Problems (VRPs) is hindered by a critical sim-to-real gap. This gap stems not only from training on oversimplified Euclidean data but also from node-based architectures incapable of handling the node-and-edge-based features with correlated asymmetric cost matrices, such as those for real-world distance and duration. We introduce RRNCO, a novel architecture specifically designed to address these complexities. RRNCO's novelty lies in two key innovations. First, its Adaptive Node Embedding (ANE) efficiently fuses spatial coordinates with real-world distance features using a learned contextual gating mechanism. Second, its Neural Adaptive Bias (NAB) is the first mechanism to jointly model asymmetric distance, duration, and directional angles, enabling it to capture complex, realistic routing constraints. Moreover, we introduce a new VRP benchmark grounded in real-world data crucial for bridging this sim-to-real gap, featuring asymmetric distance and duration matrices from 100 diverse cities, enabling the training and validation of NCO solvers on tasks that are more representative of practical settings. Experiments demonstrate that RRNCO achieves state-of-the-art performance on this benchmark, significantly advancing the practical applicability of neural solvers for real-world logistics. Our code, dataset, and pretrained models are available at https://github.com/ai4co/real-routing-nco.

## 1 Introduction

Vehicle routing problems (VRPs) are combinatorial optimization (CO) problems that represent foundational challenges in logistics and supply chain management, directly impacting operations across diverse sectors, including last-mile delivery services, disaster response management, and urban mobility. These NP-hard optimization problems require determining optimal routes for a fleet of vehicles while satisfying various operational constraints. While several traditional methods have been developed over decades (Laporte & Nobert, 1987; Vidal, 2022; Wouda et al., 2024; Perron & Furnon, 2023; Applegate et al., 2003; Wouda & Lan, 2023), these often face challenges in real-world applications. Their computational complexity makes them impractical for large-scale and real-time applications. Moreover, they often require careful parameter tuning, problem-specific adaptations, significant domain expertise, and lengthy development. With the global logistics market exceeding the 10 trillion USD mark in 2025 (Research and Markets, 2024), improvements in routing efficiency can yield substantial cost savings and environmental benefits.

Neural Combinatorial Optimization (NCO) has emerged as a promising paradigm for solving CO problems such as the VRP (Bengio et al., 2021; Wu et al., 2024a). By automatically learning heuristics directly from data, i.e., by using Reinforcement Learning (RL), NCO approaches for VRPs can potentially overcome the limitations of traditional methods by providing efficient solutions without requiring extensive domain expertise and by providing more scalable solutions (Kool et al., 2019; Zhou et al., 2023). Recent advances in NCO have demonstrated impressive results on synthetic VRP instances, suggesting the potential for learning-based approaches to achieve significant impact in real-world logistics optimization (Kwon et al., 2020; Luo et al., 2024; Ye et al., 2024b; Hottung et al., 2025).

---

*Equal contribution

However, while real-world VRPs encompass various dynamic and operational complexities, the transition from synthetic to practical applications faces a primary topological challenge.

Firstly, most existing NCO research primarily relies on simplified synthetic datasets for both training and testing that fail to capture the foundational asymmetries of real-world road networks, particularly asymmetric travel times and distances arising from road networks with diverse conditions (Osaba, 2020; Thyssens et al., 2023). Hence, a comprehensive framework for real-world data generation is needed to bridge this gap. Secondly, most current NCO architectures are based on the node-based transformer paradigm (Vaswani et al., 2017) and, as such, are not designed to effectively and efficiently embed the rich edge features and structural information present in real-world routing problems, limiting their practical applicability (Kwon et al., 2021). A new neural approach capable of effectively encoding information such as asymmetric durations and distances is thus needed to bridge the sim-to-real gap.

Our Real Routing NCO (RRNCO) bridges the critical topological gap between simplified NCO research and real-world routing applications —as illustrated in Fig. 1—through architectural innovations that specifically target the asymmetric nature of practical routing problems while serving as an extensible foundation for complex environments. We make two fundamental contributions. First, on the modeling side, we introduce a novel neural architecture with two key technical innovations: (i) an **Adaptive Node Embedding (ANE)** that dynamically fuses coordinates and distance information via learned contextual gating and probability-weighted sampling; and (ii) a **Neural Adaptive Bias (NAB)**, the first mechanism

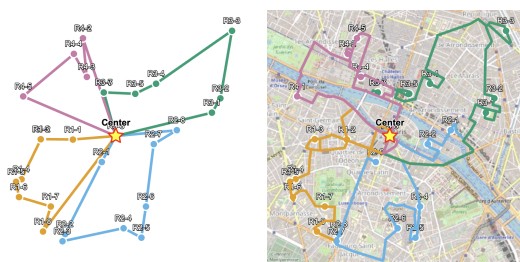

Figure 1: [Left] Most NCO works consider simplified Euclidean settings. [Right] Our work models real-world instances where durations and travel times can be asymmetric.

to jointly model asymmetric distance and duration matrices within a deep routing framework, guiding our Adaptation Attention Free Module (AAFM). To validate our approach, we construct a comprehensive benchmark dataset from 100 diverse cities, featuring real-world asymmetric distance and duration matrices from OpenStreetMap (OpenStreetMap contributors, 2025).

**Our contributions**: (1) A novel NCO architecture (RRNCO) with ANE and NAB to natively handle real-world routing asymmetries. (2) An extensive, open-source VRP dataset from 100 cities with asymmetric matrices. (3) State-of-the-art empirical results on realistic VRP instances. (4) Open-source code and data to foster reproducible research.

## 2 RELATED WORKS

**Neural Combinatorial Optimization (NCO)** Neural approaches to combinatorial optimization learn heuristics directly from data, reducing reliance on domain expertise (Bengio et al., 2021). Methods are typically classified as construction or improvement. Construction methods sequentially generate solutions, pioneered by Pointer Networks (Vinyals et al., 2015) and now led by Transformer-based autoregressive models (Kool et al., 2019; Kwon et al., 2020) for their strong ability to capture complex structures. Non-autoregressive variants predict edge-probability heatmaps in a single pass (Joshi et al., 2020), with later work enhancing performance via stronger models and search strategies (Ye et al., 2024a; Sun & Yang, 2024; Xia et al., 2024; Kim et al., 2025). Improvement methods iteratively refine an initial solution through learned operators or policies (Hottung & Tierney, 2019; Ma et al., 2023; Hottung et al., 2021; Son et al., 2023; Li et al., 2023; Chalumeau et al., 2023; Kim et al., 2021; Ma et al., 2021; Ye et al., 2024b; Zheng et al., 2024). Our work targets autoregressive construction, striking an effective balance between speed and solution quality for real-world logistics.

**Vehicle Routing Problem (VRP) Datasets** A significant gap exists between NCO research and real-world applicability, largely due to the datasets used for training and evaluation. For decades, the community has relied on established benchmarks like TSPLIB (Reinelt, 1991) and CVRPLIB

(Lima et al., 2014). While invaluable for standardization, these datasets are typically based on symmetric Euclidean distances, assuming travel costs are equal in both directions ($d_{ij} = d_{ji}$). This simplification fails to capture the inherent asymmetry of real road networks caused by one-way streets, traffic patterns, and turn restrictions (Osaba, 2020). Some recent works have attempted to create more realistic datasets (Duan et al., 2020; Ali & Saleem, 2024), but they suffer from critical limitations for NCO research: they often rely on proprietary, commercial APIs, are static and cannot be generated online (a key requirement for data-hungry RL agents), can be slow to generate, and are not always publicly released. Furthermore, they often omit crucial information like travel *durations*, which can be decoupled from distance in real traffic. Our work directly addresses these gaps by providing a fast, open-source, and scalable data generation framework that produces asymmetric distance and duration matrices from real-world city topologies.

**NCO for VRPs** The application of NCO to VRPs has evolved from early adaptations of recurrent models (Nazari et al., 2018) to the now-dominant Transformer-based encoder-decoder architectures (Kool et al., 2019; Kwon et al., 2020; Kim et al., 2022; Luo et al., 2023; Zhou et al., 2024b; Huang et al., 2025; Luo et al., 2025; Berto et al., 2025b). These models have demonstrated impressive performance but are fundamentally node-centric; their attention mechanisms operate on node embeddings, making it non-trivial to incorporate rich structural information contained in edge features like a full distance matrix. This limitation is a primary contributor to the sim-to-real gap. To address this, some works have explored encoding edge information. GCN-based approaches (Duan et al., 2020) and attention via row and column embeddings of MatNet (Kwon et al., 2021) introduced early ways to handle asymmetry, with GOAL (Drakulic et al., 2024) incorporating edge data with cross-attention. While promising, existing methods typically handle only a single cost matrix (e.g., distance) and fail to leverage the correlated modalities of real-world routing (distance, duration, geometry). Efficiently fusing multiple asymmetric edge features remains an open challenge. Our model, RRNCO, addresses this with Adaptive Node Embeddings (ANE) and a Neural Adaptive Bias (NAB) mechanism that learns a unified routing context from distance, duration, and angle.

## 3 PRELIMINARIES

### 3.1 VEHICLE ROUTING PROBLEMS

Vehicle Routing Problems (VRPs) are a class of combinatorial optimization problems that aim to find optimal routes for a fleet of vehicles serving a set of customers by minimizing a cost function. The simplest variant, the Traveling Salesman Problem (TSP), involves finding a minimum-cost Hamiltonian cycle in a complete graph $G = (V, E)$ with $n = |V|$ locations. Real-world applications typically extend this basic formulation with operational constraints such as vehicle capacity limits (CVRP) or time windows for service (VRPTW) (Vidal et al., 2014). These problems are characterized by their input structure, consisting of node and edge features. Node features typically include location coordinates and customer demands, while edge features capture the relationships between locations. In real-world settings, these relationships are represented by distance and duration matrices, $D, T \in \mathbb{R}^{n \times n}$, where $d_{ij}$ and $t_{ij}$ denote the distance and travel time from location $i$ to $j$, respectively. Previous NCO approaches typically rely on Euclidean distances computed directly from location coordinates, avoiding the use of distance matrices entirely. While this simplification works for synthetic problems with symmetric distances ($d_{ij} = d_{ji}$, $t_{ij} = t_{ji}$), real-world instances are inherently asymmetric due to factors such as traffic patterns and road network constraints. This asymmetry, combined with the problem's combinatorial nature, presents unique challenges for learning-based approaches, particularly in effectively encoding and processing the rich structural information present in edge features.

### 3.2 SOLVING VRPs WITH GENERATIVE MODELS

VRPs can be solved as a sequential decision process, and deep generative models can be learned to efficiently do so (Wu et al., 2024b). Given a problem instance $x$ containing both node features (such as coordinates, demands, and time windows) and edge features (distance and duration matrices $D, T$), we construct solutions through autoregressive generation. Our model iteratively selects the next location to visit based on the current partial route until all locations are covered. This construction process naturally aligns with how routes are executed in practice and allows the model to main-

tain feasibility constraints throughout generation. In this work, we consider the encoder-decoder framework as in Kool et al. (2019). Formally, let $\theta = \{\theta_f, \theta_g\}$ denote the combined parameters of our encoder and decoder networks. We learn a policy $\pi_\theta$ that maps input instances to solutions through:

$$\boldsymbol{h} = f_{\theta_f}(\boldsymbol{x}), \tag{1a}$$

$$\pi_\theta(\boldsymbol{a}|\boldsymbol{x}) = \prod_{t=1}^{T} g_{\theta_g}(a_t|a_{t-1}, \ldots, a_1, \boldsymbol{h}), \tag{1b}$$

where $\boldsymbol{h}$ represents the learned latent problem embedding, $a_t$ is the location selected at step $t$, and $a_{t-1}, \ldots, a_1$, denotes the partial route constructed so far. The architecture choices for the encoding process (Eq. (1a)) and decoding (Eq. (1b)) via $f$ and $g$, respectively, are paramount to ensure high solution quality.

### 3.3 TRAINING VIA REINFORCEMENT LEARNING

We frame the learning problem in a reinforcement learning (RL) context, which enables data-free optimization by automatically discovering heuristics. The policy parameters $\theta$ are optimized to maximize the expected reward:

$$\max_\theta J(\theta) = \mathbb{E}_{\boldsymbol{x} \sim \mathcal{D}} \mathbb{E}_{\boldsymbol{a} \sim \pi_\theta(\cdot|\boldsymbol{x})}[R(\boldsymbol{a}, \boldsymbol{x})], \tag{2}$$

where $\mathcal{D}$ is the problem distribution we sample and $R(\boldsymbol{a}, \boldsymbol{x})$ is the reward (i.e., the negative cost) of a solution. Policy gradient methods can be employed to solve this problem, such as REINFORCE with the variance-reducing POMO baseline (Kwon et al., 2020). Due to RL's exploratory, i.e., trial and error, nature, many samples are required. Thus, efficient generation and sampling of problem instances $\boldsymbol{x}$ is essential to ensure training efficiency.

## 4 REAL-WORLD ROUTING MODEL

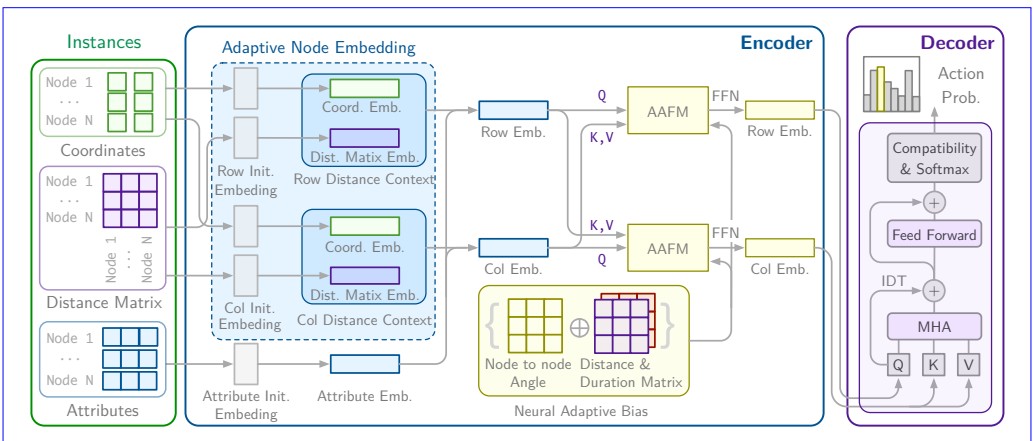

Figure 2: Our proposed RRNCO model for real-world routing.

Our model addresses real-world routing challenges where conventional methods struggle with asymmetric attributes like travel times and distances. We introduce two key innovations: (1) Adaptive Node Embedding with probability-weighted distance sampling - efficiently integrating spatial coordinates with asymmetric distances through learned contextual gating, avoiding full distance matrix processing while preserving asymmetric relationships, and (2) Neural Adaptive Bias (NAB) - the first learnable mechanism to jointly model asymmetric distance and duration matrices in deep routing architectures, replacing hand-crafted heuristics in AAFM (Zhou et al., 2024a) with data-driven contextual biases. The model uses an encoder-decoder architecture where the encoder builds comprehensive node representations and the decoder generates solutions sequentially, with our contributions focusing on enhancing the encoder's real-world routing capabilities while maintaining efficiency.

## 4.1 ENCODER

### 4.1.1 ADAPTIVE NODE EMBEDDING

The Adaptive Node Embedding module synthesizes distance-related features with node characteristics to create comprehensive node representations. A key aspect of our approach is effectively integrating two complementary spatial features: distance matrix information and coordinate-based relationships. For distance matrix information, we employ a selective sampling strategy that captures the most relevant node relationships while maintaining computational efficiency. Given a distance matrix $\mathbf{D} \in \mathbb{R}^{N \times N}$, we sample $k$ nodes for each node $i$ according to probabilities inversely proportional to their distances:

$$p_{ij} = \frac{1/d_{ij}}{\sum_{j=1}^{N} 1/d_{ij}} \tag{3}$$

where $d_{ij}$ represents the distance between nodes $i$ and $j$. The sampled distances are then transformed into an embedding space through a learned linear projection:

$$\mathbf{f}_{\text{dist}} = \text{Linear}(\mathbf{d}_{\text{sampled}}) \tag{4}$$

Coordinate information is processed separately to capture geometric relationships between nodes. For each node, we first compute its spatial features based on raw coordinates. These features are then projected into the same embedding space through another learned linear transformation:

$$\mathbf{f}_{\text{coord}} = \text{Linear}(\mathbf{x}_{\text{coord}}) \tag{5}$$

To effectively combine these complementary spatial representations, we employ a Contextual Gating mechanism:

$$\mathbf{h} = \mathbf{g} \odot \mathbf{f}_{\text{coord}} + (1 - \mathbf{g}) \odot \mathbf{f}_{\text{dist}} \tag{6}$$

where $\odot$ is the Hadamard product and $\mathbf{g}$ represents learned gating weights determined by a multi-layer perceptron (MLP) :

$$\mathbf{g} = \sigma(\text{MLP}([\mathbf{f}_{\text{coord}}; \mathbf{f}_{\text{dist}}])) \tag{7}$$

This gating mechanism allows the model to adaptively weigh the importance of coordinate-based and distance-based features for each node, enabling more nuanced spatial representation. To handle asymmetric routing scenarios effectively, we follow the approach introduced in (Kwon et al., 2021) and generate dual embeddings for each node: row embeddings $\mathbf{h}^r$ and column embeddings $\mathbf{h}^c$. These embeddings are then combined with other node characteristics (such as demand or time windows) through learned linear transformations to produce the combined node representations:

$$\mathbf{h}_{\text{comb}}^r = \text{MLP}([\mathbf{h}^r; \mathbf{f}_{\text{node}}]) \tag{8}$$

$$\mathbf{h}_{\text{comb}}^c = \text{MLP}([\mathbf{h}^c; \mathbf{f}_{\text{node}}]) \tag{9}$$

where $\mathbf{f}_{\text{node}}$ represents additional node features such as demand or time windows, which are transformed by an additional linear layer. This dual embedding approach allows the RRNCO model to better capture and process asymmetric relationships in real-world routing scenarios.

### 4.1.2 NEURAL ADAPTIVE BIAS FOR AAFM

Having established comprehensive node representations through our adaptive embedding approach, RRNCO employs an Adaptation Attention-Free Module (AAFM) based on Zhou et al. (2024a) to model complex inter-node relationships. The AAFM operates on the dual representations $\mathbf{h}_{\text{comb}}^r$ and $\mathbf{h}_{\text{comb}}^c$ to capture asymmetric routing patterns through our novel Neural Adaptive Bias (NAB) mechanism. The AAFM operation is defined as:

$$\text{AAFM}(Q, K, V, A) = \sigma(Q) \odot \frac{\exp(A) \cdot (\exp(K) \odot V)}{\exp(A) \cdot \exp(K)} \tag{10}$$

where $Q = \mathbf{W}^Q \mathbf{h}_{\text{comb}}^r$, $K = \mathbf{W}^K \mathbf{h}_{\text{comb}}^c$, $V = \mathbf{W}^V \mathbf{h}_{\text{comb}}^c$, with learnable weight matrices $\mathbf{W}^Q$, $\mathbf{W}^K$, $\mathbf{W}^V$. While Zhou et al. (2024a) defines the adaptation bias $A$ heuristically as $-\alpha \cdot \log(N) \cdot d_{ij}$ (with learnable $\alpha$, node count $N$, and distance $d_{ij}$), we introduce a Neural Adaptive Bias (NAB) that learns asymmetric relationships directly from data. NAB processes distance matrix $\mathbf{D}$, angle matrix $\mathbf{\Phi}$ with entries $\phi_{ij} = \arctan2(y_j - y_i, x_j - x_i)$, and optionally duration matrix $\mathbf{T}$, enabling joint

modeling of spatial-temporal asymmetries inherent in real-world routing. Let $\mathbf{W}_D, \mathbf{W}_\Phi, \mathbf{W}_T \in \mathbb{R}^E$:

$$\mathbf{D}_{emb} = \text{ReLU}(\mathbf{D}\mathbf{W}_D)\mathbf{W}'_D \tag{11}$$

$$\mathbf{\Phi}_{emb} = \text{ReLU}(\mathbf{\Phi}\mathbf{W}_\Phi)\mathbf{W}'_\Phi \tag{12}$$

$$\mathbf{T}_{emb} = \text{ReLU}(\mathbf{T}\mathbf{W}_T)\mathbf{W}'_T \tag{13}$$

We then apply contextual gating to fuse these heterogeneous information sources. When duration information is available, we employ a multi-channel gating mechanism with softmax normalization:

$$\mathbf{G} = \text{softmax}\left(\frac{[\mathbf{D}_{\text{emb}};\ \mathbf{\Phi}_{\text{emb}};\ \mathbf{T}_{\text{emb}}]\mathbf{W}_G}{\exp(\tau)}\right) \tag{14}$$

where $[\mathbf{D}_{emb}; \mathbf{\Phi}_{emb}; \mathbf{T}_{emb}] \in \mathbb{R}^{B \times N \times N \times 3E}$ is the concatenation of all embeddings, $\mathbf{W}_G \in \mathbb{R}^{3E \times 3}$ is a learnable weight matrix, and $\tau$ is a learnable temperature parameter. The fused representation is computed as:

$$\mathbf{H} = \mathbf{G}_1 \odot \mathbf{D}_{\text{emb}} + \mathbf{G}_2 \odot \mathbf{\Phi}_{\text{emb}} + \mathbf{G}_3 \odot \mathbf{T}_{\text{emb}} \tag{15}$$

Finally, the adaptive bias matrix $\mathbf{A}$ is obtained by projecting the fused embedding $\mathbf{H}$ to a scalar value:

$$\mathbf{A} = \mathbf{H}\mathbf{w}_{out} \in \mathbb{R}^{B \times N \times N} \tag{16}$$

where $\mathbf{w}_{out} \in \mathbb{R}^E$ is a learnable weight vector. The resulting $\mathbf{A}$ matrix serves as a learned inductive bias that captures complex asymmetric relationships arising from the interplay between distances, directional angles, and travel durations. The Neural Adaptive Bias is then incorporated into the Adaptation Attention Free Module (AAFM). Specifically, we employ the operation defined in (10) by replacing the generic matrix $A$ with the adaptive matrix generated by NAB.

The Neural Adaptive Bias (NAB), applied through the Adaptation Attention Free Module (AAFM), yields final node representations $h_F^r$ and $h_F^c$ after $l$ passes through AAFM. These representations result from RRNCO's encoding process, leveraging joint modeling of distance, angle, and duration to capture complex asymmetric patterns in real-world routing networks.

## 4.2 DECODER

### 4.2.1 DECODER ARCHITECTURE

The decoder architecture integrates key elements from ReLD (Huang et al., 2025) and Mat-Net (Kwon et al., 2021) to autoregressively construct solutions using the dense node embeddings produced by the encoder. At each decoding step $t$, it takes as input the row and column node embeddings alongside a context vector that encapsulates the current partial solution state, such as the last visited node and dynamic attributes like remaining capacity. This context serves as the query in a multi-head attention mechanism to aggregate information from the embeddings, followed by residual connections and a multi-layer perceptron to refine the query vector. The resulting query is then employed in a compatibility layer to compute selection probabilities for feasible nodes, incorporating a negative logarithmic distance heuristic to prioritize nearby options and enhance exploration efficiency. This design enables our model to dynamically adapt static embeddings to evolving contexts, yielding strong performance across vehicle routing problems; for full technical details, including equations and implementation specifics, please refer to the Appendix A.

## 5 REAL-WORLD VRP DATASET

A significant challenge in applying Neural Combinatorial Optimization (NCO) to real-world routing is the lack of realistic datasets. Most existing benchmarks rely on synthetic instances with symmetric, Euclidean distances, failing to capture the complexities of actual road networks, such as one-way streets and traffic-dependent travel times, which lead to asymmetric distance and duration matrices. To bridge this gap, we introduce a new, large-scale dataset for real-world VRPs. We developed a comprehensive data generation pipeline that leverages the OpenStreetMap Routing Engine (OSRM) (OpenStreetMap contributors, 2025) to create detailed topological maps for 100 diverse cities worldwide. Each map includes location coordinates along with their corresponding asymmetric distance

and duration matrices. Furthermore, we designed an efficient online subsampling method to generate a virtually unlimited number of VRP instances for training our reinforcement learning agent. This approach ensures that our model is trained on data that faithfully represents real-world routing challenges. In addition to serving as a benchmark, the dataset provides a structured basis for evaluating NCO solvers under realistic conditions and helps narrow the simulation-to-real gap, offering a useful resource for future research on practical logistics. Details of the city selection, data generation framework, and subsampling methodology are given in Appendix B.

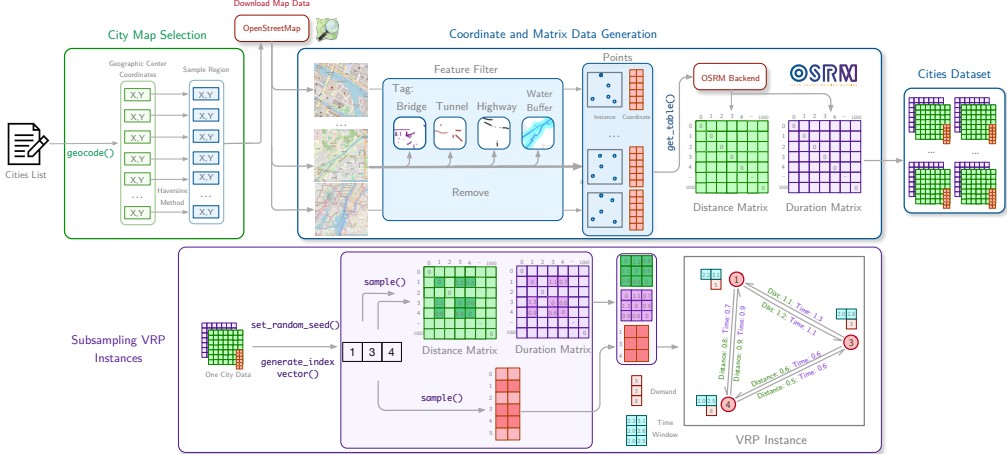

Figure 3: Overview of our RRNCO real-world data generation and sampling framework. We generate a dataset of real-world cities with coordinates and respective distance and duration matrices obtained via OSRM. Then, we efficiently subsample instances as a set of coordinates and their matrices from the city map dataset with additional generated VRP features.

# 6 EXPERIMENTS

## 6.1 EXPERIMENTAL SETUP

**Classical Baselines** In the experiments, we compare three SOTA traditional optimization approaches: LKH3 (Helsgaun, 2017): a heuristic algorithm with strong performance on (A)TSP problems, PyVRP (Wouda et al., 2024): a specialized solver for VRPs with comprehensive constraint handling capabilities; and Google OR-Tools (Perron & Didier, 2024): a versatile optimization library for CO problems.

**Learning-Based Methods** We compare against SOTA NCO methods divided in two categories. *1) Node-only encoding learning methods*: POMO (Kwon et al., 2020), an end-to-end multi-trajectory RL-based method based on attention mechanisms; MTPOMO (Liu et al., 2024), a multi-task variant of POMO; MVMoE (Zhou et al., 2024b), a mixture-of-experts variant of MTPOMO; RF (Berto et al., 2025b): an RL-based foundation model for VRPs; ELG (Gao et al., 2024), a hybrid of local and global policies for routing problems; BQ-NCO (Drakulic et al., 2023): a decoder-only transformer trained with supervised learning; LEHD (Luo et al., 2023): a supervised learning-based heavy decoder model and AAFM (Zhou et al., 2024a), introduced in the ICAM framework as an attention-free alternative enabling instance-conditioned adaptation. *2) Node and edge encoding learning methods*: GCN (Duan et al., 2020): a graph convolutional network with encoding of edge information for routing; MatNet (Kwon et al., 2021): an RL-based solver encoding edge features via matrices; ReLD-MTL and ReLD-MoEL (Huang et al., 2025), which incorporate identity mapping and feed-forward decoder refinements to significantly improve cross-size and cross-problem generalization and GOAL (Drakulic et al., 2024): a generalist agent trained via supervised learning for several CO problems, including routing problems.

**Training Configuration** We perform training runs under the same settings for fair comparison for our model, MatNet for ATSP and ACVRP, and GCN for ACVRP. Node-only models do not

Table 1: Performance comparison across real-world routing tasks and distributions. We report costs and gaps calculated with respect to best-known solutions (∗) from traditional solvers. Horizontal lines separate traditional solvers, node-only methods, node-and-edge methods, and our RRNCO. Lower is better (↓).

| Task | Method | In-distribution | | | Out-of-distribution (city) | | | Out-of-distribution (cluster) | | |
|---|---|---|---|---|---|---|---|---|---|---|
| | | Cost | Gap (%) | Time | Cost | Gap (%) | Time | Cost | Gap (%) | Time |
| ATSP | LKH3 | 38.387 | ∗ | 1.6h | 38.903 | ∗ | 1.6h | 12.170 | ∗ | 1.6h |
| | OR-Tools | 39.685 | 3.381 | 7h | 40.165 | 3.244 | 7h | 12.711 | 4.445 | 7h |
| | POMO | 51.512 | 34.192 | 10s | 50.594 | 30.051 | 10s | 30.051 | 146.926 | 10s |
| | ELG | 51.046 | 32.976 | 42s | 50.133 | 28.866 | 42s | 23.017 | 89.131 | 42s |
| | BQ-NCO | 55.933 | 45.708 | 25s | 54.739 | 40.706 | 25s | 27.872 | 129.022 | 25s |
| | LEHD | 56.099 | 46.140 | 13s | 54.811 | 40.891 | 13s | 27.819 | 128.587 | 13s |
| | MatNet | 39.915 | 3.981 | 27s | 40.548 | 4.228 | 27s | 12.886 | 5.883 | 27s |
| | GOAL | 41.976 | 9.350 | 91s | 42.590 | 9.477 | 91s | 13.654 | 12.194 | 91s |
| | AAFM | 45.992 | 19.812 | 151s | 46.588 | 19.755 | 151s | 15.211 | 24.987 | 151s |
| | RRNCO | 39.078 | 1.799 | 23s | 39.785 | 2.268 | 23s | 12.444 | 2.250 | 23s |
| ACVRP | PyVRP | 69.739 | ∗ | 7h | 70.488 | ∗ | 7h | 22.553 | ∗ | 7h |
| | OR-Tools | 72.597 | 4.097 | 7h | 73.286 | 3.969 | 7h | 23.576 | 4.538 | 7h |
| | POMO | 85.888 | 23.156 | 16s | 85.771 | 21.682 | 16s | 34.179 | 51.549 | 16s |
| | MTPOMO | 86.521 | 24.063 | 16s | 86.446 | 22.640 | 16s | 34.287 | 52.029 | 16s |
| | MVMoE | 86.248 | 23.672 | 22s | 86.111 | 22.164 | 22s | 34.135 | 51.356 | 22s |
| | RF | 86.289 | 23.731 | 17s | 86.261 | 22.377 | 16s | 34.273 | 51.967 | 16s |
| | ELG | 85.951 | 23.247 | 67s | 85.741 | 21.639 | 66s | 34.027 | 50.873 | 67s |
| | BQ-NCO | 93.075 | 33.462 | 30s | 92.467 | 31.181 | 30s | 40.110 | 77.848 | 30s |
| | LEHD | 93.648 | 34.284 | 17s | 93.195 | 32.214 | 17s | 40.048 | 77.573 | 17s |
| | ReLD-MTL | 88.331 | 26.659 | 16s | 88.037 | 24.896 | 16s | 36.169 | 60.373 | 16s |
| | ReLD-MoEL | 88.154 | 26.406 | 16s | 87.764 | 24.509 | 16s | 36.137 | 60.231 | 16s |
| | GCN | 90.546 | 29.836 | 17s | 90.805 | 28.823 | 17s | 34.417 | 52.605 | 17s |
| | MatNet | 74.801 | 7.258 | 30s | 75.722 | 7.425 | 30s | 24.844 | 10.158 | 30s |
| | GOAL | 84.341 | 20.938 | 104s | 84.097 | 19.307 | 104s | 34.318 | 52.166 | 104s |
| | AAFM | 76.663 | 9.928 | 11s | 77.811 | 10.389 | 11s | 25.131 | 11.431 | 11s |
| | RRNCO | 72.145 | 3.450 | 26s | 73.010 | 3.578 | 26s | 23.282 | 3.231 | 26s |
| ACVRPTW | PyVRP | 118.056 | ∗ | 7h | 118.513 | ∗ | 7h | 39.253 | ∗ | 7h |
| | OR-Tools | 119.681 | 1.377 | 7h | 120.147 | 1.379 | 7h | 39.903 | 1.655 | 7h |
| | POMO | 132.883 | 12.559 | 18s | 132.743 | 12.007 | 17s | 50.503 | 28.661 | 18s |
| | MTPOMO | 133.135 | 12.773 | 17s | 132.921 | 12.158 | 18s | 50.372 | 28.328 | 18s |
| | MVMoE | 132.871 | 12.549 | 24s | 132.700 | 11.971 | 23s | 50.333 | 28.227 | 24s |
| | RF | 132.887 | 12.563 | 18s | 132.731 | 11.997 | 18s | 50.422 | 28.455 | 18s |
| | ReLD-MTL | 132.722 | 12.423 | 18s | 132.856 | 12.102 | 18s | 51.680 | 31.659 | 18s |
| | ReLD-MoEL | 132.594 | 12.314 | 18s | 132.621 | 11.904 | 18s | 51.647 | 31.575 | 18s |
| | GOAL | 134.699 | 14.098 | 107s | 135.001 | 13.912 | 107s | 47.966 | 22.197 | 107s |
| | RRNCO | 122.466 | 3.736 | 52s | 122.986 | 3.775 | 52s | 40.937 | 4.290 | 52s |

necessitate retraining since our datasets are already normalized in the $[0, 1]^2$ coordinates ranges (with locations sampled uniformly), and we do not retrain supervised-learning models since they would necessitate labeled data. The model is trained for about 24 hours on $4\times$ NVIDIA A100 40GB GPUs, with all training settings and train, test dataset details provided in Appendix Section C.1 and Appendix Section B.4.

**Testing Protocol** The test data consists of in-distribution evaluation for 1) *In-dist*: new instances generated from the 80 cities seen during training, 2) *OOD (city)* out-of-distribution generalization over new city maps and 3) *OOD (cluster)* out-of-distribution generalization to new location distributions across maps. The test batch size is 32, and a data augmentation factor of 8 is applied to all models except supervised learning-based ones, i.e., LEHD, BQ-NCO, and GOAL. All evaluations are conducted on an NVIDIA A6000 GPU paired with an Intel(R) Xeon(R) CPU @ 2.20GHz[1].

---

[1]Code: https://github.com/ai4co/real-routing-nco

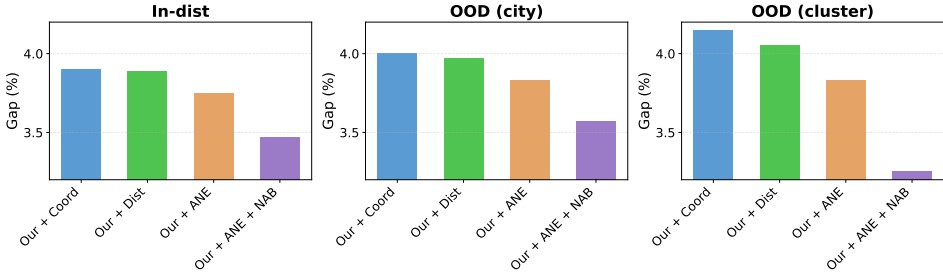

Figure 4: Study of our proposed model with different initial contexts: coordinates, distances, Adaptive Node Embedding (ANE), and Neural Adaptive Bias (NAB). ANE and NAB perform best, particularly in out-of-distribution (OOD) cases.

## 6.2 MAIN RESULTS

Table 1 shows the results between our and the baseline models across ATSP, ACVRP, and ACVRPTW tasks, with inference times in parentheses. The results clearly demonstrate that our method achieves state-of-the-art performance across all problem settings, consistently outperforming existing neural solvers in both solution quality and computational efficiency. Notably, unlike previous approaches that require separate models for different problem types, our method effectively handles all routing problems within a single unified framework. This key advantage highlights the model's adaptability and scalability across diverse problem instances while maintaining strong generalization for both in-distribution and out-of-distribution scenarios in real-world settings.

## 6.3 ANALYSES

**Ablation study on proposed components** We perform an ablation study on proposed model components in Fig. 4: initial contexts with coordinates, distances, and our Adaptive Node Embedding(ANE), as well as the Neural Adaptive Bias (NAB). We find ANE and NAB perform the best, particularly in out-of-distribution (OOD) cases. Remarkably, in cluster distributions, the NAB shows a relative improvement greater than $15\%$.

**Granular ablation on NAB inputs** To further investigate the contribution of each input modality to the Neural Adaptive Bias (NAB), we conduct a granular ablation study on ACVRPTW in Table 2. Starting from the full model that jointly uses distance, duration, and angle matrices, we progressively remove components. The results show that each modality contributes to the final performance: removing duration increases the gap from 3.74% to 3.93% (in-distribution), and further removing angle increases it to 4.06%. The performance degradation is more pronounced in the OOD (cluster) setting, where the gap increases from 4.30% to 4.84%, demonstrating that the joint modeling of all three features is essential for robust generalization.

Table 2: Granular ablation on NAB inputs for ACVRPTW. We progressively remove duration and angle from the full NAB model.

| Model | In-distribution | | OOD (city) | | OOD (cluster) | |
|---|---|---|---|---|---|---|
| | Cost | Gap (%) | Cost | Gap (%) | Cost | Gap (%) |
| PyVRP | 118.056 | ∗ | 118.513 | ∗ | 39.253 | ∗ |
| RRNCO Full (D + T + Φ) | **122.467** | **3.74** | **122.986** | **3.78** | **40.937** | **4.29** |
| − Duration (D + Φ) | 122.693 | 3.93 | 123.249 | 4.00 | 41.077 | 4.65 |
| − Duration − Angle (D only) | 122.849 | 4.06 | 123.364 | 4.09 | 41.151 | 4.84 |

**Importance of real-world data generators** We study the effect of training different models on different data generators, including the ATSP one from MatNet (Kwon et al., 2021), adding random

Table 3: Comparison of routing solvers and their training data generators on  real-world data.

| Method | Data Gen. | In-dist | | OOD City | | OOD Clust. | |
|---|---|---|---|---|---|---|---|
| | | Cost | Gap% | Cost | Gap% | Cost | Gap% |
| LKH3 | – | 38.39 | ∗ | 38.90 | ∗ | 12.17 | ∗ |
| MatNet | ATSP | 80.86 | 110.70 | 81.04 | 108.30 | 27.78 | 128.23 |
| RRNCO | Noise | 41.35 | 7.72 | 42.01 | 7.98 | 13.66 | 12.20 |
| MatNet | Real | 39.92 | 3.98 | 40.55 | 4.23 | 12.89 | 5.88 |
| RRNCO | Real | **39.08** | **1.80** | **39.79** | **2.27** | **12.44** | **2.25** |

noise to break symmetries in distance matrices, and our proposed real-world generator when testing in the real world. Table 3 demonstrates our proposed real-world data generation achieves remarkable improvements in both in-distribution and out-of-distribution settings.

**Generalization to stochastic VRPs**   RRNCO demonstrates remarkable performance when applied to real-world topologies that surpass prior works.  We further evaluate the robustness of RRNCO and generalizability to new settings, i.e., under stochastic and time-dependent traffic conditions and conduct experiments on the recently released Stochastic Multi-period Time-dependent VRP (SMTVRP) benchmark from SVRPBench (Heakl et al., 2025).  We also benchmark against additional traditional baselines, including Nearest Neighbor + 2-opt (NN+2opt) (Laporte & Nobert, 1987; Potvin & Rousseau, 1995) and Ant Colony Optimization (ACO) (Dorigo et al., 2006).

As shown in Table 4, RRNCO achieves the lowest cost (601.03) while maintaining full feasibility and zero time window violations. Notably, OR-Tools fails to find feasible solutions for 75.8% of instances within the time limit, highlighting the practical advantage of neural solvers in time-constrained scenarios. These results demonstrate that RRNCO generalizes effectively to more complex, time-dependent routing problems beyond the benchmarks used for training. For more details of the dataset, baselines, and training configurations, please refer to Appendix E.

Table 4: Performance comparison on SMTVRP benchmark. Feasibility indicates the proportion of instances with valid solutions. Lower cost is better.

| Method | Cost | Feasibility | Runtime | TW Violations |
|---|---|---|---|---|
| ACO | 763.52 | 1.00 | 41.3s | 0 |
| OR-Tools | 610.08 | 0.242 | 1000s | 45.15 |
| NN + 2opt | 969.96 | 1.00 | 10s | 0 |
| RF | 602.20 | 1.00 | 0.05s | 0 |
| GOAL | 1319.00 | 1.00 | 0.28s | 0 |
| RRNCO | **601.03** | 1.00 | 0.20s | 0 |

## 7  CONCLUSION

In this paper, we introduced RRNCO, a novel Neural Combinatorial Optimization architecture bridging simplified benchmarks and real-world routing challenges. Our core contribution is a model explicitly handling asymmetric and multi-modal travel costs through two key innovations: Adaptive Node Embedding (ANE) efficiently fusing coordinates with sampled distance features, and a Neural Adaptive Bias (NAB) mechanism. This NAB represents the first approach to jointly model multiple asymmetric metrics—distance, duration, and directional angles—in real-world routing problems. To validate our model, we constructed a large-scale dataset with realistic routing data from 100 diverse cities. This dataset provides a reproducible and diverse testbed that supports future research on robust and generalizable NCO solvers. On this challenging benchmark, RRNCO achieves state-of-the-art performance among NCO methods. By releasing our model and dataset, we aim to accelerate progress in practical, deployable neural optimization solutions.

## REPRODUCIBILITY STATEMENT

To facilitate reproducibility of our results, we provide complete access to our implementation, including dataset generators and training configurations. The source code is available at https://github.com/ai4co/real-routing-nco. We will additionally disclose the URL of generated data and model weights on HuggingFace. Detailed descriptions of model architectures, dataset creation, training procedures, and experimental setups are provided in the main paper and appendix.

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

APPENDIX

## A    DETAILED MODEL DECODER ARCHITECTURE

The decoder architecture combines key elements from the ReLD and MatNet to effectively process the dense node embeddings generated by the encoder and construct solutions for vehicle routing problems. At each decoding step $t$, the decoder takes as input the row and column node embeddings $(h_F^r, h_F^c)$ produced by the encoder and a context vector $h_c$. The composition of $h_c$ adapts to the problem type. For VRP variants (ACVRP and ACVRPTW), the context relies on the last visited node and the dynamic state. In contrast, for ATSP, we additionally include the embedding of the first node $a_0$ to explicitly anchor the tour's origin. Formally, the context vector is defined as:

$$h_c = \begin{cases} [h_{a_0}^r, h_{a_{t-1}}^r, D^t] \in \mathbb{R}^{2d_h + d_{\text{attr}}} & \text{for ATSP,} \\ [h_{a_{t-1}}^r, D^t] \in \mathbb{R}^{d_h + d_{\text{attr}}} & \text{for ACVRP and ACVRPTW.} \end{cases} \quad (17)$$

where $D^t \in \mathbb{R}^{d_{\text{attr}}}$ represents the dynamic features derived from the state $s^t$. The specific contents of $D^t$ are tailored to the constraints of each problem variant: for ACVRP, $D^t$ consists of the **current load**; for ACVRPTW, it includes both the **current load** and the **current time**. In the case of ATSP, which is unconstrained by capacity or time windows, no dynamic features are utilized (i.e., $D^t = \emptyset$). To aggregate information from the node embeddings, the decoder applies a multi-head attention (MHA) mechanism, using the context vector $h_c$ as the query and $H^t \in \mathbb{R}^{|F^t| \times d_h}$ as the key and value:

$$h_c' = \text{MHA}(h_c, W^{key}h_F^c, W^{val}h_F^c). \quad (18)$$

The ReLD model introduces a direct influence of context by adding a residual connection between the context vector $h_c$ and the refined query vector $h_c'$:

$$h_c' = h_c' + \text{IDT}(h_c), \quad (19)$$

where $\text{IDT}(\cdot)$ is an identity mapping function that reshapes the context vector to match the dimension of the query vector, allowing context-aware information to be directly embedded into the representation. To further enhance the decoder performance, an MLP with residual connections is incorporated to introduce non-linearity into the computation of the final query vector $q_c$:

$$q_c = h_c' + \text{MLP}(h_c'). \quad (20)$$

The MLP consists of two linear transformations with a ReLU activation function, transforming the decoder into a transformer block with a single query that can model complex relationships and adapt the embeddings based on the context. Finally, the probability $p_i$ of selecting node $i \in F^t$ is calculated by applying a compatibility layer with a negative logarithmic distance heuristic score:

$$p_i = \left[ \text{Softmax} \left( C \cdot \tanh \left( \frac{(q_c)^T W^\ell h_F^c}{\sqrt{d_h}} - \log(\text{dist}_i) \right) \right) \right]_i \quad (21)$$

where $C$ is a clipping hyperparameter, $d_h$ is the embedding dimension, and $\text{dist}_i$ denotes the distance between node $i$ and the last selected node $a_{t-1}$. This heuristic guides the model to prioritize nearby nodes during the solution construction process. The combination of ReLD's architectural modifications and MatNet's decoding mechanism with our rich, learned encoding enables the RRNCO model to effectively leverage static node embeddings while dynamically adapting to the current context, leading to improved performance on various vehicle routing problems.

## B    REAL-WORLD VRP DATASET GENERATION

Existing methodologies often require integrating massive raw datasets (e.g., traffic simulators and multi-source spatial data) – for instance, Barauskas et al. (2023) rely on simplistic synthetic benchmarks, which are either resource-intensive or lack real-world complexity (Gunawan et al., 2021).

To address these limitations, we design a three-step pipeline to create a diverse and realistic vehicle routing dataset aimed at training and testing NCO models. First, we select cities worldwide based on

multi-dimensional urban descriptors (morphology, traffic flow regimes, land-use mix). Second, we develop a framework using the Open Source Routing Machine (OSRM) (Luxen & Vetter, 2011) to create city maps with topological data, generating both precise location coordinates and their corresponding distance and duration matrices between each other. Finally, we efficiently subsample these topologies to generate diverse VRP instances by adding routing-specific features such as demands and time windows, thus preserving the inherent spatial relationships while enabling the rapid generation of instances with varying operational constraints, leveraging the precomputed distance/duration matrices from the base maps. The whole pipeline is illustrated in Fig. 3.

## B.1 CITY MAP SELECTION

We select a list of 100 cities distributed across six continents, with 25 in Asia, 21 in Europe, 15 each in North America and South America, 14 in Africa, and 10 in Oceania. The selection emphasizes urban diversity through multiple dimensions, including population scale (50 large cities >1M inhabitants, 30 medium cities 100K-1M, and 20 small cities <100K), infrastructure development stages, and urban planning approaches. Cities feature various layouts, from grid-based systems like Manhattan to radial patterns like Paris and organic developments like Fez, representing different geographic and climatic contexts from coastal to mountain locations. We prioritized cities with reliable data availability while balancing between globally recognized metropolitan areas and lesser-known urban centers, providing a comprehensive foundation for evaluating vehicle routing algorithms under diverse real-world conditions. Moreover, by including cities from developing regions, we aim to advance transportation optimization research that could benefit underprivileged areas and contribute to their socioeconomic development.

## B.2 TOPOLOGICAL DATA GENERATION FRAMEWORK

In the second stage, we generate base maps that capture real urban complexities. This topological data generation is composed itself of three key components: geographic boundary information, point sampling from road networks, and travel information computation.

**Geographic boundary information**    We establish standardized 9 km$^2$ areas (3×3 km) centered on each target city's municipal coordinates, ensuring the same spatial coverage across different urban environments. Given that the same physical distance corresponds to different longitudinal spans at different latitudes due to the Earth's spherical geometry, we need a precise distance calculation method: thus, the spatial boundaries are computed using the Haversine spherical distance formulation (Chopde & Nichat, 2013):

$$d = 2R \cdot \arcsin\left(\sqrt{\sin^2\left(\frac{\Delta\phi}{2}\right) + \cos(\phi_1)\cos(\phi_2)\sin^2\left(\frac{\Delta\lambda}{2}\right)}\right) \tag{22}$$

where $d$ is the distance between two points along the great circle, $R$ is Earth's radius (approximately 6,371 kilometers), $\phi_1$ and $\phi_2$ are the latitudes of point 1 and point 2 in radians, $\Delta\phi = \phi_2 - \phi_1$ represents the difference in latitudes, and $\Delta\lambda = \lambda_2 - \lambda_1$ represents the difference in longitudes. This enables precise spatial boundary calculations and standardized cross-city comparisons while maintaining consistent analysis areas across different geographic locations.

**Point sampling from road networks**    Our RRNCO framework interfaces with OpenStreetMap (OpenStreetMap contributors, 2025) for point sampling. More specifically, we extract both road networks and water features within defined boundaries using `graph_from_bbox` and `features_from_bbox`[2]. Employing boolean indexing, the sampling process implements several filtering mechanisms to filter the DataFrame and ensure point quality: we exclude bridges, tunnels, and highways to focus on accessible street-level locations and create buffer zones around water features to prevent sampling from (close to) inaccessible areas. Points are then generated through a weighted random sampling approach, where road segments are weighted by their length to ensure uniform spatial distribution.

---

[2]https://osmnx.readthedocs.io/en/stable/user-reference.html

**Travel information computation** The travel information computation component leverages a locally hosted Open Source Routing Machine (OSRM) server (Luxen & Vetter, 2011) to calculate real travel distances and durations between sampled points, ensuring full reproducibility of results. Through the efficient `get_table` function in our router implementation via the OSRM table service[3], we can process a complete 1000x1000 origin-destination matrix within 18 seconds, making it highly scalable for urban-scale analyses. In contrast to commercial API-based approaches that require more than 20 seconds for 350×350 matrices (Ali & Saleem, 2024), our open-source local OSRM implementation achieves the same computations in approximately 5 seconds. Additionally, it enables the rapid generation of multiple instances from small datasets with negligible computational cost per iteration epoch. The RRNCO framework finally processes this routing data through a normalization strategy that addresses both unreachable destinations and abnormal travel times. This step captures real-world routing complexities, including one-way streets, turn restrictions, and varying road conditions, resulting in asymmetric distance and duration matrices that reflect actual urban travel patterns. All computations are performed locally[4], allowing for consistent results and independent verification of the analysis pipeline.

## B.3 VRP Instance Subsampling

From the large-scale city base maps, we generate diverse VRP instances by subsampling a set of locations along with their corresponding distance and duration matrices, allowing us to generate an effectively unlimited number of instances while preserving the underlying structure. The subsampling process follows another three-step procedure:

1. *Index Selection*: Given a city dataset containing $N_{\text{tot}}$ locations, we define a subset size $N_{\text{sub}}$ representing the number of locations to be sampled for the VRP instance. We generate an index vector $\mathbf{s} = (s_1, s_2, \ldots, s_{N_{\text{sub}}})$ where each $s_i$ is drawn from $\{1, \ldots, N_{\text{tot}}\}$, ensuring unique selections. 2. *Matrix Subsampling*: Using $\mathbf{s}$, we extract submatrices from the precomputed distance matrix $D \in \mathbb{R}^{N_{\text{tot}} \times N_{\text{tot}}}$ and duration matrix $T \in \mathbb{R}^{N_{\text{tot}} \times N_{\text{tot}}}$, forming instance-specific matrices $D_{\text{sub}} = D[\mathbf{s}, \mathbf{s}] \in \mathbb{R}^{N_{\text{sub}} \times N_{\text{sub}}}$ and $T_{\text{sub}} = T[\mathbf{s}, \mathbf{s}] \in \mathbb{R}^{N_{\text{sub}} \times N_{\text{sub}}}$, preserving spatial relationships among selected locations. 3. *Feature Generation*: Each VRP can have different features. For example, in Asymmetric Capacitated VRP (ACVRP) we can generate a demand vector $\mathbf{d} \in \mathbb{R}^{N_{\text{sub}} \times 1}$, such that $\mathbf{d} = (d_1, d_2, \ldots, d_{N_{\text{sub}}})^\top$, where each $d_i$ represents the demand at location $s_i$. Similarly, we can extend to ACVRPTW (time windows) represented as $\mathbf{W} \in \mathbb{R}^{N_{\text{sub}} \times 2}$, where $\mathbf{W} = \{(w_1^{\text{start}}, w_1^{\text{end}}), \ldots, (w_{N_{\text{sub}}}^{\text{start}}, w_{N_{\text{sub}}}^{\text{end}})\}$, defining the valid service interval for each node.

Unlike previous methods that generate static datasets offline (Duan et al., 2020; Ali & Saleem, 2024), our RRNCO generation framework dynamically generates instances on the fly in few milliseconds, reducing disk memory consumption while maintaining high diversity. Fig. 3 illustrates the overall process, showing how a city map is subsampled using an index vector to create VRP instances with distance and duration matrices enriched with node-specific features such as demands and time windows. Our approach allows us to generate a (arbitrarily) large number of problem instances from a relatively small set of base topology maps totaling around 1.5GB, in contrast to previous works that required hundreds of gigabytes of data to produce just a few thousand instances.

## B.4 Additional Data Information

We present a comprehensive urban mobility dataset encompassing 100 cities across diverse geographical regions worldwide. For each city, we collected 1000 sampling points distributed throughout the same size urban area. The dataset includes the precise geographical coordinates (latitude and longitude) for each sampling point. Additionally, we computed and stored complete distance and travel time matrices between all pairs of points within each city, resulting in 1000×1000 matrices per city. The cities in our dataset exhibit significant variety in their characteristics, including population size (ranging from small to large), urban layout patterns (such as grid, organic, mixed, and historical layouts), and distinct geographic features (coastal, mountain, river, valley, etc.). The dataset covers multiple regions including Asia, Oceania, Americas, Europe, and Africa. This diversity in urban

---

[3] https://project-osrm.org/docs/v5.24.0/api/#table-service

[4] Our framework can also be extended to include real-time commercial map API integrations and powerful traffic forecasting to obtain better-informed routing (Wang & Xu, 2011; Qu & Wu, 2025), which we leave as future works.

environments enables comprehensive analysis of mobility patterns across different urban contexts and geographical settings. Table 5 on the following page provides information about our topology dataset choices.

Table 5: Comprehensive City Details

| City | Population | Layout | Geographic Features | Region | Split |
|---|---|---|---|---|---|
| Addis Ababa | Large | Organic | Highland | East Africa | Train |
| Alexandria | Large | Mixed | Coastal | North Africa | Train |
| Amsterdam | Large | Canal grid | River | Western Europe | Train |
| Almaty | Large | Grid | Mountain | Central Asia | Train |
| Asunción | Medium | Grid | River | South America | Test |
| Athens | Large | Mixed | Historical | Southern Europe | Train |
| Auckland | Large | Harbor layout | Isthmus | Oceania | Train |
| Baku | Large | Mixed | Coastal | Western Asia | Train |
| Bangkok | Large | River layout | River | Southeast Asia | Train |
| Barcelona | Large | Grid & historic | Coastal | Southern Europe | Train |
| Beijing | Large | Ring layout | Plains | East Asia | Train |
| Bergen | Small | Fjord | Coastal mountain | Northern Europe | Train |
| Brisbane | Large | River grid | River | Oceania | Train |
| Buenos Aires | Large | Grid | River | South America | Train |
| Bukhara | Small | Medieval | Historical | Central Asia | Test |
| Cape Town | Large | Mixed colonial | Coastal&mountain | Southern Africa | Train |
| Cartagena | Medium | Colonial | Coastal | South America | Train |
| Casablanca | Large | Mixed colonial | Coastal | North Africa | Train |
| Chengdu | Large | Grid | Basin | East Asia | Train |
| Colombo | Medium | Colonial grid | Coastal | South Asia | Train |
| Chicago | Large | Grid | Lake | North America | Test |
| Christchurch | Medium | Grid | Coastal plain | Oceania | Train |
| Copenhagen | Large | Mixed | Coastal | Northern Europe | Train |
| Curitiba | Large | Grid | Highland | South America | Train |
| Cusco | Medium | Historic mixed | Mountain | South America | Test |
| Daejeon | Large | Grid | Valley | East Asia | Train |
| Dakar | Medium | Peninsula grid | Coastal | West Africa | Train |
| Dar es Salaam | Large | Coastal grid | Coastal | East Africa | Train |
| Denver | Large | Grid | Mountain | North America | Train |
| Dhaka | Large | Organic | River | South Asia | Train |
| Dubai | Large | Linear modern | Coastal& desert | Western Asia | Train |

*Continued on next page*

Table 5 – *Continued from previous page*

| City | Population | Layout | Geographic Features | Region | Split |
|------|-----------|--------|---------------------|--------|-------|
| Dublin | Large | Georgian grid | Coastal | Northern Europe | Train |
| Dubrovnik | Small | Medieval walled | Coastal | Southern Europe | Train |
| Edinburgh | Medium | Historic mixed | Hills | Northern Europe | Train |
| Fez | Medium | Medieval organic | Historical | North Africa | Test |
| Guatemala City | Large | Valley grid | Valley | Central America | Train |
| Hanoi | Large | Mixed | River | Southeast Asia | Train |
| Havana | Large | Colonial | Coastal | Caribbean | Train |
| Helsinki | Large | Grid | Peninsula | Northern Europe | Train |
| Hobart | Small | Mountain harbor | Harbor | Oceania | Test |
| Hong Kong | Large | Vertical | Harbor | East Asia | Train |
| Istanbul | Large | Mixed | Strait | Western Asia | Train |
| Kigali | Medium | Hill organic | Highland | East Africa | Train |
| Kinshasa | Large | Organic | River | Central Africa | Train |
| Kuala Lumpur | Large | Modern mixed | Valley | Southeast Asia | Test |
| Kyoto | Large | Historical grid | Valley | East Asia | Train |
| La Paz | Large | Valley organic | Mountain | South America | Train |
| Lagos | Large | Organic | Coastal | West Africa | Train |
| Lima | Large | Mixed grid | Coastal desert | South America | Train |
| London | Large | Radial organic | River | Northern Europe | Test |
| Los Angeles | Large | Grid sprawl | Coastal basin | North America | Train |
| Luanda | Large | Mixed | Coastal | Southern Africa | Train |
| Mandalay | Large | Grid | River | Southeast Asia | Train |
| Marrakech | Medium | Medina | Desert edge | North Africa | Train |
| Medellín | Large | Valley grid | Mountain | South America | Train |
| Melbourne | Large | Grid | River | Oceania | Train |
| Mexico City | Large | Mixed | Valley | North America | Test |
| Montevideo | Large | Grid | Coastal | South America | Train |
| Montreal | Large | Mixed | Island | North America | Train |
| Moscow | Large | Ring layout | River | Eastern Europe | Train |
| Mumbai | Large | Linear coastal | Coastal | South Asia | Test |
| Nairobi | Large | Mixed | Highland | East Africa | Train |

*Continued on next page*

Table 5 – *Continued from previous page*

| City | Population | Layout | Geographic Features | Region | Split |
|------|-----------|--------|--------------------|--------|-------|
| New Orleans | Medium | Colonial | River delta | North America | Train |
| New York City | Large | Grid | Coastal | North America | Train |
| Nouméa | Small | Peninsula | Coastal | Oceania | Test |
| Osaka | Large | Grid | Harbor | East Asia | Test |
| Panama City | Large | Coastal modern | Coastal | Central America | Train |
| Paris | Large | Radial | River | Western Europe | Train |
| Perth | Large | Coastal sprawl | Coastal | Oceania | Test |
| Port Moresby | Medium | Harbor sprawl | Coastal hills | Oceania | Train |
| Porto | Medium | Medieval organic | River mouth | Southern Europe | Train |
| Prague | Large | Historic grid | River | Central Europe | Train |
| Quebec City | Medium | Historic walled | River | North America | Test |
| Quito | Large | Linear valley | Highland | South America | Test |
| Reykjavik | Small | Modern grid | Coastal | Northern Europe | Test |
| Rio de Janeiro | Large | Coastal organic | Mountain& coastal | South America | Train |
| Rome | Large | Historical organic | Seven hills | Southern Europe | Test |
| Salvador | Large | Mixed historic | Coastal | South America | Train |
| Salzburg | Small | Medieval core | River | Central Europe | Train |
| San Francisco | Large | Hill grid | Peninsula | North America | Train |
| San Juan | Medium | Mixed historic | Coastal | Caribbean | Test |
| Santiago | Large | Grid | Valley | South America | Train |
| São Paulo | Large | Sprawl | Highland | South America | Train |
| Seoul | Large | Mixed | River | East Asia | Train |
| Shanghai | Large | Modern mixed | River | East Asia | Train |
| Singapore | Large | Planned | Island | Southeast Asia | Train |
| Stockholm | Large | Archipelago | Island | Northern Europe | Train |
| Sydney | Large | Harbor organic | Harbor | Oceania | Train |
| Taipei | Large | Grid | Basin | East Asia | Train |
| Thimphu | Small | Valley organic | Mountain | South Asia | Train |
| Tokyo | Large | Mixed | Harbor | East Asia | Test |
| Toronto | Large | Grid | Lake | North America | Train |
| Ulaanbaatar | Large | Grid | Valley | East Asia | Train |
| Valparaíso | Medium | Hill organic | Coastal hills | South America | Train |

Table 5 – *Continued from previous page*

| City | Population | Layout | Geographic Features | Region | Split |
|------|-----------|--------|---------------------|--------|-------|
| Vancouver | Large | Grid | Peninsula | North America | Train |
| Vienna | Large | Ring layout | River | Central Europe | Train |
| Vientiane | Medium | Mixed | River | Southeast Asia | Train |
| Wellington | Medium | Harbor basin | Coastal hills | Oceania | Train |
| Windhoek | Small | Grid | Highland | Southern Africa | Test |
| Yogyakarta | Medium | Traditional | Cultural center | Southeast Asia | Train |

## C  ADDITIONAL EXPERIMENTAL DETAILS

### C.1  HYPERPARAMETER DETAILS

Table 6 shows the hyperparameters we employ for RRNCO. The configuration can be changed through `yaml` files as outlined in RL4CO (Berto et al., 2025a), which we employ as the base framework for our codebase.

Table 6: Hyperparameters for RRNCO.

| Hyperparameter | Value |
|----------------|-------|
| Optimizer | Adam |
| Learning Rate | $4 \times 10^{-4}$ |
| LR Decay Schedule | 0.1 at epochs 180, 195 |
| Batch Size | 256 |
| Instances per Epoch | 100,000 |
| Embedding Dimension | 128 |
| Feedforward Dimension | 512 |
| AAFM Layers | 12 |
| Clipping $C$ | 10 |

### C.2  TESTING DATASET

For each dataset of the main experiments Section 6.2, we use subsampling as described in Section B.3 and sample 1,280 instances for each test set. Whole test sets and seeds are provided in the shared code for reproducibility.

### C.3  BASELINES DETAILS

All evaluations are conducted on an NVIDIA A6000 GPU paired with an Intel(R) Xeon(R) CPU @ 2.20GHz. For neural methods, we employ the provided models and code in the original repositories to ensure fairness and reproducibilty.

We evaluate all traditional solvers on a single CPU core sequentially. To reflect a realistic deployment scenario, we do not perform instance-specific hyperparameter tuning, instead relying on the robust default configurations provided by each library. For LKH-3 (Helsgaun, 2017), we utilize the standard parameter set from the official distribution (e.g., `PATCHING_C=3`, `PATCHING_A=2` for VRPs), imposing a `TIME_LIMIT` of 5 seconds per instance. For PyVRP (Wouda et al., 2024), we employ the default Hybrid Genetic Search (HGS) parameters specifically `nb_elite=4` and `generation_size=40` with a 20-second time limit via `MaxRuntime`. Similarly, for Google OR-Tools (Perron & Didier, 2024), we configure the routing model to use the

`GUIDED_LOCAL_SEARCH` metaheuristic initialized with `PATH_CHEAPEST_ARC`, restricted to a 20-second budget. This protocol, consistent with Liu et al. (2024), results in a total evaluation time of approximately 7 hours for the test set (1,280 instances).

# D    USE OF LARGE LANGUAGE MODELS

Large language models were employed solely as general-purpose writing assistants. Their use was restricted to refining phrasing, improving clarity, and correcting grammar in draft versions of the manuscript. All research ideas, methodologies, analyses, results, and interpretations were conceived, executed, and validated exclusively by the authors. Any text generated with the assistance of LLMs was thoroughly reviewed, edited, and integrated by the authors to ensure accuracy, correctness, and compliance with academic standards.

# E    ADDITIONAL MATERIAL ON STOCHASTIC BENCHMARKS

We further evaluate RRNCO on routing scenarios with time-varying travel conditions. Specifically, we benchmark on the Stochastic Multi-period Time-dependent VRP (SMTVRP), which incorporates realistic traffic dynamics following the simulation protocol of SVRPBench (Heakl et al., 2025).

## E.1    BENCHMARK SETUP

**Dynamic Travel Time Model**    Real-world travel times exhibit significant temporal variation due to congestion patterns, stochastic delays, and unexpected incidents. We model the travel time from node $a$ to $b$ departing at time $t$ as:

$$T(a, b, t) = T_{\text{base}}(a, b) + T_{\text{congestion}}(a, b, t) + T_{\text{incident}}(t), \tag{23}$$

where $T_{\text{base}}(a, b) = D(a, b)/V$ represents the free-flow travel time based on Euclidean distance $D(a, b)$ and average speed $V$. The congestion component captures systematic daily patterns:

$$T_{\text{congestion}}(a, b, t) = B(a, b, t) \cdot R(t), \tag{24}$$

with $B(a, b, t)$ encoding deterministic congestion and $R(t)$ introducing stochastic variability.

The congestion factor combines temporal and spatial dependencies:

$$B(a, b, t) = \alpha \cdot \underbrace{\left[\beta + \gamma \sum_{p \in \{\text{am}, \text{pm}\}} \mathcal{G}(t; \mu_p, \sigma_p)\right]}_{F_{\text{time}}(t)} \cdot \underbrace{\left(1 - e^{-D(a,b)/\lambda}\right)}_{F_{\text{dist}}(D)}, \tag{25}$$

where $\mathcal{G}(t; \mu, \sigma)$ denotes a Gaussian kernel. The bimodal temporal structure with peaks at $\mu_{\text{am}} = 8$ and $\mu_{\text{pm}} = 17$ reflects typical morning and evening rush hours. The distance factor captures the empirical observation that longer trips have higher congestion exposure.

To model traffic variability, $R(t)$ follows a log-normal distribution with time-dependent parameters:

$$R(t) \sim \text{LogNormal}(\mu_R(t), \sigma_R(t)), \tag{26}$$

where both $\mu_R(t)$ and $\sigma_R(t)$ increase during peak hours to reflect heightened uncertainty.

Incident-induced delays are modeled as:

$$T_{\text{incident}}(t) = \mathbb{1}[\text{incident at } t] \cdot \Delta_{\text{incident}}, \tag{27}$$

where incident occurrences follow a time-inhomogeneous Poisson process with elevated rates during nighttime hours ($\mu_{\text{night}} = 21$), and delay durations $\Delta_{\text{incident}} \sim U(0.5, 2.0)$ hours align with industry clearance statistics.

**Time Window Generation**  Customer availability windows are sampled to reflect realistic delivery scenarios. We distinguish between two customer types with distinct temporal preferences:

$$T_{\text{start}}^{(\text{res})} \sim 0.5 \cdot \mathcal{N}(\mu_{\text{am}}, \sigma_{\text{am}}^2) + 0.5 \cdot \mathcal{N}(\mu_{\text{pm}}, \sigma_{\text{pm}}^2), \tag{28}$$

for residential customers with morning ($\mu_{\text{am}} = 480$ min) and evening ($\mu_{\text{pm}} = 1140$ min) availability peaks, and

$$T_{\text{start}}^{(\text{com})} \sim \mathcal{N}(\mu_{\text{biz}}, \sigma_{\text{biz}}^2), \tag{29}$$

for commercial customers centered on business hours ($\mu_{\text{biz}} = 780$ min). Window durations are sampled uniformly within problem-specific bounds. Complete parameter specifications follow SVRP-Bench (Heakl et al., 2025).

### E.2  BASELINE CONFIGURATIONS

We evaluate a set of baseline methods on the stochastic multi-period time-dependent VRP (SMTVRP) benchmark to ensure rigorous and fair comparisons. All methods are configured following standard practices in SVRPBench and prior VRP literature, with adjustments only when necessary to accommodate time-dependent travel information.

**ACO (Ant Colony Optimization)**  We adopt a conventional parameterization commonly used in the SVRP literature. To balance solution quality and computational efficiency in dynamic environments, we set the number of ants to 50 and the maximum number of iterations to 100. The pheromone-related parameters are fixed to $\alpha = 1.0$, $\beta = 2.0$, and evaporation rate $\rho = 0.5$. At each temporal update in SMTVRP, heuristic costs are recomputed while pheromone trails are preserved to maintain stability across time periods.

**OR-Tools**  We use the guided local search (GLS) implementation included in SVRPBench with a strict time limit of 1000 seconds per instance. This extended budget allows the solver to explore large neighborhoods under dynamic updates. Following each change in the time-dependent duration matrix, OR-Tools is restarted from the most recent feasible solution when possible, but no custom tuning beyond the default GLS configuration is introduced.

**NN + 2-opt**  In accordance with the SVRPBench protocol, a Nearest Neighbor (NN) heuristic is first used to construct an initial solution, followed by a 2-opt local improvement phase. The search terminates upon convergence or after reaching a maximum runtime of 10 seconds per instance. Under dynamic conditions, NN + 2-opt is rerun at every update step to preserve consistency with the updated travel-time information.

**GOAL**  We evaluate GOAL using its official pre-trained checkpoint trained on CVRPTW instances. No additional training or adaptation is performed. For each temporal update, GOAL re-evaluates its autoregressive decoding process under the current duration matrix, thereby testing its zero-shot generalization capacity to dynamic inputs.

**RouteFinder**  We initialize RouteFinder from its official pre-trained model and apply Efficient Active Learning (EAL) for 20 epochs to adapt the policy to the SMTVRP distributions. After EAL adaptation, inference proceeds in a closed-loop manner: at each step, the updated travel-time matrix is processed, and actions are generated based on the current state of the dynamic environment.

### E.3  MULTI-SNAPSHOT NAB FOR STOCHASTIC ROUTING

To extend RRNCO to robustly handle time-dependent and stochastic traffic variations, we introduce a Multi-Snapshot variant of the Neural Adaptive Bias (NAB). While the standard NAB fuses a single duration matrix, real-world traffic is highly dynamic and uncertain. To address this, we integrate a dynamics-informed traffic simulation directly into the encoder pipeline and extend the NAB architecture to process multiple temporal "snapshots" of traffic conditions simultaneously.

### E.3.1 DYNAMICS-INFORMED INPUT GENERATION

To approximate the stochastic nature of real-world travel times, we employ the dynamics-informed traffic modeling framework proposed in SVRPBench (Heakl et al., 2025). Instead of a single static duration matrix, the model takes as input a set of $K$ distinct duration snapshots $\{\mathbf{T}_1, \ldots, \mathbf{T}_K\}$. In our experiments, we set $K = 3$ to represent distinct traffic regimes (e.g., morning peak, noon, evening peak).

These snapshots are generated using Gaussian kernels centered at learnable "time anchors" to model congestion peaks, combined with stochastic noise injection. Specifically, for a snapshot $k$, the duration $t_{ij}^{(k)}$ is derived by modulating the base travel time with a congestion factor drawn from a time-dependent Gaussian distribution, multiplicative noise sampled from a LogNormal distribution to simulate flow variability, and sparse additive noise modeled via a Poisson process to account for random incidents (accidents). This ensures the model receives a global, "look-ahead" view of potential traffic states and their associated uncertainties.

### E.3.2 MULTI-SNAPSHOT GATING MECHANISM

We extend the Neural Adaptive Bias (NAB) module to aggregate these heterogeneous temporal views. The Multi-Snapshot NAB employs a unified gating network that dynamically weighs the importance of the static spatial structure against the variable traffic conditions.

First, we project the static Distance matrix $\mathbf{D}$, the Relative Angle matrix $\mathbf{\Phi}$, and each of the $K$ dynamic Duration snapshots into a shared high-dimensional embedding space:

$$\mathbf{D}_{\text{emb}} = \text{ReLU}(\mathbf{D}\mathbf{W}_D)\mathbf{W}'_D, \tag{30}$$

$$\mathbf{\Phi}_{\text{emb}} = \text{ReLU}(\mathbf{\Phi}\mathbf{W}_\Phi)\mathbf{W}'_\Phi, \tag{31}$$

$$\mathbf{T}_{\text{emb}}^{(k)} = \text{ReLU}(\mathbf{T}_k\mathbf{W}_T)\mathbf{W}'_T, \quad \forall k \in \{1, \ldots, K\}. \tag{32}$$

To fuse these $K+2$ feature maps, we compute a set of scalar importance weights via a learned gating network. Let $\mathbf{E}_{\text{concat}} = [\mathbf{D}_{\text{emb}}; \mathbf{\Phi}_{\text{emb}}; \mathbf{T}_{\text{emb}}^{(1)}; \ldots; \mathbf{T}_{\text{emb}}^{(K)}]$ denote the concatenation of all embeddings. We compute the gating weights $\mathbf{g} \in \mathbb{R}^{2+K}$ via a softmax function:

$$\mathbf{g} = \text{softmax}\left(\frac{\mathbf{E}_{\text{concat}}\mathbf{W}_G}{\exp(\tau)}\right), \tag{33}$$

where $\mathbf{W}_G$ is a learnable weight matrix and $\tau$ is the temperature parameter. The final fused representation $\mathbf{H}_{\text{fused}}$ is computed as a probability-weighted sum of the static and dynamic features:

$$\mathbf{H}_{\text{fused}} = g_1 \odot \mathbf{D}_{\text{emb}} + g_2 \odot \mathbf{\Phi}_{\text{emb}} + \sum_{k=1}^{K} g_{k+2} \odot \mathbf{T}_{\text{emb}}^{(k)}. \tag{34}$$

Similar to the standard NAB, this fused representation is projected to a scalar bias matrix $\mathbf{A} = \mathbf{H}_{\text{fused}}\mathbf{w}_{\text{out}}$. This bias $\mathbf{A}$ modulates the attention scores in the Adaptation Attention Free Module (AAFM), guiding the solver to avoid routes that are consistently congested across the simulated snapshots or to leverage time windows where traffic is predicted to be lighter.

### E.4 TRAINING DETAILS

**RRNCO (NAB)** The standard RRNCO model equipped with the Neural Adaptive Bias (NAB) is trained for 100 epochs on static instances sampled from the real-world generator introduced in the main paper. Each epoch consists of 100,000 training instances, with a batch size of 256 and the Adam optimizer with a learning rate of $4 \times 10^{-4}$. The learning rate is decayed following the schedule described in Table 6 of the appendix. No fine-tuning is performed on the SMTVRP benchmark; instead, the model is directly evaluated in a dynamic closed-loop setting.

**RRNCO (Multi-Snapshot NAB)** As detailed in Section E.3, this variant integrates temporal variability by augmenting the Neural Adaptive Bias (NAB) module with a dynamics-informed traffic simulator and a temporal gating mechanism designed to statically fuse multiple duration snapshots.

In our experiments, the model receives $K = 3$ temporal duration snapshots at training time, corresponding to short-horizon predictions of traffic fluctuations. The model is trained for 100 epochs under the same reinforcement learning setup as the base RRNCO. During dynamic evaluation on SMTVRP, the NAB module processes and fuses the $K$ snapshots once during the initial encoding step. This process enables the resultant node representations to embed a global context of potential traffic variations, allowing the autoregressive decoder to anticipate near-future congestion patterns without requiring dynamic re-encoding at every decision step.

**Inference Protocol** Both RRNCO variants operate autoregressively in a closed-loop dynamic environment. After each routing action, the updated time-dependent duration matrix is provided by the SMTVRP simulator. The model recomputes its route continuation based on this updated information. A decoding augmentation factor of 8 is applied during inference to reduce variance and enhance route stability across dynamic updates.

### E.5 RESULTS FOR MULTI-SNAPSHOT NAB

Table 7 demonstrates that RRNCO with Multi-Snapshot NAB achieves state-of-the-art performance with a cost of 594.19, surpassing both the standard NAB (601.03) and RouteFinder (602.20). This improvement confirms that fusing multiple temporal snapshots allows the model to better anticipate traffic stochasticity. On the other hand, OR-Tools fails to find feasible solutions for 75.8% of instances despite a 1000s budget. RRNCO maintains 100% feasibility and zero time window violations, offering superior robustness and speed (0.20s) with virtually no additional inference latency compared to the standard model.

Table 7: Performance comparison on SMTVRP benchmark. Feasibility indicates the proportion of instances with valid solutions. Lower cost is better.

| Method | Cost | Feasibility | Runtime | TW Violations |
|---|---|---|---|---|
| ACO | 763.52 | 1.00 | 41.3s | 0 |
| OR-Tools | 610.08 | 0.242 | 1000s | 45.15 |
| NN + 2opt | 969.96 | 1.00 | 10s | 0 |
| RF | 602.20 | 1.00 | 0.05s | 0 |
| GOAL | 1319.00 | 1.00 | 0.28s | 0 |
| RRNCO (NAB) | 601.03 | 1.00 | 0.20s | 0 |
| RRNCO (Multi-Snapshot NAB) | **594.19** | 1.00 | 0.20s | 0 |

