# OpenReview forum: "RRNCO: Towards Real-World Routing with Neural Combinatorial Optimization"
_ICLR.cc/2026/Conference — ICLR 2026 Poster_

### Official Review · Reviewer_XMP3 · 2025-10-31

**Soundness:** 3
**Presentation:** 4
**Contribution:** 4
**Rating:** 8
**Confidence:** 3

**Summary:**

This paper investigates the application of neural combinatorial optimization solvers to real-world problem instances. Specifically, the authors use a adaptive attention-free module from prior work to incorporate real-world travel time and distance features, in contrast to existing studies that rely on synthetic data with Euclidean distances. They further compile a comprehensive dataset covering problem instances from 100 cities, which serves as a testbed for evaluating the proposed method against established baselines.

**Strengths:**

I personally found this paper very exciting, as it presents (to the best of my knowledge) the first large-scale real-world dataset for benchmarking neural combinatorial solvers. This represents a valuable step forward, and the dataset has the potential to benefit many researchers in the community.

**Weaknesses:**

I do not have much to complain about this paper, below is a question that I am curious about.

- Can the adaptive attention-free module handle features other than travel time and distances? What about contextual features that are shared among node pairs, for example, time of the day and weather?

**Questions:**

See above

---

> ### Author Response · Authors · 2025-12-03
> **Official Response to Reviewer XMP3**
>
> We sincerely thank the reviewer for the enthusiastic and encouraging feedback. We are delighted that you found our paper exciting and recognized the significant value of our contributions, specifically the introduction of the first large-scale real-world dataset for benchmarking neural combinatorial solvers. We agree that bridging the sim-to-real gap is critical for the community, and we are glad you see our work as a valuable step forward in this direction.
>
> Below, we address your question regarding the extensibility of our architecture.
>
> ---
>
>
>
> **Q1. Can the adaptive attention-free module handle features other than travel time and distances? What about contextual features that are shared among node pairs, for example, time of the day and weather?**
>
> Thank you for this insightful question. The answer is **yes**, our proposed architecture, specifically the Neural Adaptive Bias (NAB) mechanism which feeds into the Adaptation Attention Free Module (AAFM), is natively designed to support and fuse multi-modal pairwise inputs, including contextual features like time of day or weather conditions.
>
> The core design philosophy of the NAB is to project heterogeneous inputs into a shared embedding space and then fuse them via a learned gating mechanism. As detailed in Equation 14 and 15 of the revised manuscript, the gating mechanism currently computes weights for distance, angle, and duration. Extending this to include additional contextual features is straightforward and follows the same architectural logic:
>
> 1.  **Embedding New Modalities:** For a new feature type, such as weather conditions (e.g., precipitation levels affecting road friction) or time of day (e.g., global traffic congestion factors), we would first process the raw input through a dedicated linear projection or MLP to map it into the same dimension as our current embeddings. For global features like "time of day," this information can be broadcast across all node pairs or combined with edge-specific features.
> 2.  **Extended Gating Mechanism:** We would then extend the softmax gating function (Eq. 14) to include an additional channel for this new embedding. The model would automatically learn a corresponding weight matrix for this channel.
> 3.  **Learnable Importance:** A key advantage of this design is that we avoid manual feature engineering. The learnable contextual gating allows the model to dynamically determine the relative importance of these new features. For instance, the model could learn to weigh "weather" features heavily during heavy rain scenarios while prioritizing "duration" or "distance" in standard conditions.
>
> In fact, we have demonstrated this capability in our rebuttal revision. To address stochastic and time-dependent routing, we introduced a **Multi-Snapshot NAB** (detailed in **Appendix Section E.3**). In this extension, rather than a single duration matrix, the model takes multiple "snapshots" of traffic conditions representing different time periods. The NAB successfully fuses these additional inputs via the gating mechanism to predict robust routes. This confirms that the architecture is flexible enough to handle various additional dense feature matrices or contextual inputs without structural changes to the core framework.

---

### Official Review · Reviewer_mJYV · 2025-10-31

**Soundness:** 2
**Presentation:** 2
**Contribution:** 3
**Rating:** 4
**Confidence:** 4

**Summary:**

This paper introduces RRNCO, a new neural combinatorial optimization (NCO) architecture targeting practical real-world vehicle routing problems (VRP). The paper also presents a new and comprehensive VRP dataset based on OpenStreetMap data from 100 cities, enabling both in-distribution and out-of-distribution evaluation.

**Strengths:**

1. The paper introduces a large, scalable, and diverse real-world VRP dataset that captures the essential complexities (asymmetry, real-world travel times) often ignored in current research. Making this dataset open-source will prove highly valuable to the research community.

2. The proposed ANE and NAB mechanisms are well-motivated, addressing the need to handle node-and-edge features natively, especially asymmetric matrices, which are more representative in real logistics.

3. This paper is clearly written.

**Weaknesses:**

1. Problem Definition: The definition of “real-world VRP” in the paper appears to focus almost exclusively on the presence of asymmetric cost matrices (distance, duration). While this is a major source of realism, real-world VRPs are also characterized by other features such as dynamic distance changes, rich operational constraints (e.g., skills, priorities, real-time updates), time-dependent travel times, etc. The current problem definition could be more comprehensive and better justified in context.

2. Comparison to OR-Tools in ATSP: In the ATSP experiments, the baseline comparisons do not include Google OR-Tools.

3. Computation Time Reporting: For some baselines (e.g., OR-Tools takes “several hours”), the exact computation protocol is unclear.

4. Scope of Generalization: Experiments are done on the proposed real-world dataset with varied geographies, but it’s not clear if the model's performance holds for even more complex cases. Better to show results on estinblished dataset or benchmarks as well.

**Questions:**

1. On the Definition of "Real-World": Besides asymmetry, what other real-world features (as per logistics practice) does your approach not cover? How might your model adapt to additional complexities such as time-dependent or dynamic edge lengths?

2. On ATSP Baselines: Why was OR-Tools not included in the ATSP comparison, given that it supports ATSP solutions? Are there technical limitations, or was this a deliberate choice? Please elaborate.


3. On OR-Tools Computation Time: The reported computation time for OR-Tools on VRP benchmarks is shown as “7h” or “several hours.” How was this measured?

4. On Generalization to Other VRP Benchmarks: How easily can your RRNCO be adapted to incorporate other benchmarks?

---

> ### Author Response · Authors · 2025-12-03
> **Official Response to Reviewer mJYV (Part1)**
>
> We thank the reviewer for their insightful comments and for recognizing the value of our work. We are particularly grateful for the appreciation of our large scale open source dataset, the motivation behind our ANE and NAB mechanisms, and the clarity of our presentation. We have carefully addressed the concerns raised and conducted additional experiments to demonstrate the robustness and extensibility of RRNCO.
>
> ---
>
> **1. Problem Definition and Scope of "Real-World" VRPs**
>
> We appreciate the reviewer’s comment regarding the definition of "real-world VRP." We agree that the term encompasses a broad spectrum of complexities, including dynamic updates, rich operational constraints, and time-dependent factors.
>
> * **Focus on Asymmetry:** Our work primarily targets the "Sim-to-Real" gap caused by the fundamental topology of road networks. The shift from symmetric Euclidean distances (standard in NCO literature) to asymmetric cost matrices is the most immediate and critical hurdle preventing neural solvers from functioning in actual logistics. We prioritized asymmetry because, without solving this, no amount of constraint handling makes a solver viable for real roads.
> * **Modularity and Extensibility:** We explicitly designed the RRNCO architecture to be modular. The Adaptive Node Embedding (ANE) and Neural Adaptive Bias (NAB) are designed to be extensible to other "real-world" features. As we discuss in our response to your question on generalization (Question 4), we have successfully extended the NAB mechanism to handle time-dependent traffic conditions without altering the core backbone of the model.
> * **Revisions:** We have revised Section 1 and the problem definition to more precisely scope our contribution. We now explicitly state that while we focus on the fundamental topological gap (asymmetry), our architecture is designed as a foundation to support future extensions into dynamic and highly constrained environments.
>
> **2. Comparison to OR Tools in ATSP**
>
> We sincerely appreciate the reviewer noting the absence of Google OR Tools in our initial ATSP analysis. We recognize that OR Tools is a critical industrial baseline for verifying practical applicability. To address this, we have conducted a comprehensive evaluation of OR Tools on our ATSP benchmarks, covering In distribution, Out of distribution (City), and Out of distribution (Cluster) settings. The results confirm that RRNCO maintains its state of the art performance and significant efficiency advantages against this strong baseline.
>
> * **Performance Comparison:** As detailed in the updated results below, RRNCO consistently outperforms OR Tools across all data distributions.
>     * **In Distribution:** RRNCO achieves a gap of **1.797%**, whereas OR Tools results in a gap of **3.381%**.
>     * **Generalization (OOD Cluster):** The performance difference becomes even more pronounced in the challenging OOD Cluster setting, which tests the model on unseen node distributions. While OR Tools degrades to a **4.445%** gap, RRNCO demonstrates superior robustness with a **2.301%** gap. This highlights that our learned heuristics generalize better to novel spatial structures than the local search heuristics employed by OR Tools under the same time constraints.
>
> * **Computational Efficiency:** RRNCO is orders of magnitude faster. While OR Tools requires approximately **7 hours** to solve the full test set (due to sequential CPU execution typical of traditional solvers), RRNCO completes the entire evaluation in just **22 seconds** on GPU. This speed is critical for real time logistics where routing decisions must be made near instantaneously.
>
> We have updated Table 1 in the revised manuscript to include these results. A summary of the key metrics is provided below:
>
> | Task | Method | In Dist Gap (%) | OOD (City) Gap (%) | OOD (Cluster) Gap (%) | Time |
> | :--- | :--- | :---: | :---: | :---: | :---: |
> | **ATSP** | LKH3 (Oracle) | 0.000 | 0.000 | 0.000 | 1.6h |
> | | **OR Tools** | 3.381 | 3.244 | 4.445 | 7h |
> | | MatNet | 3.981 | 4.228 | 5.883 | 27s |
> | | **RRNCO (Ours)** | **1.797** | **2.262** | **2.301** | **22s** |

---

> ### Author Response · Authors · 2025-12-03
> **Official Response to Reviewer mJYV (Part2)**
>
> **3. Clarification on Computation Time Reporting**
>
> We understand the reviewer's concern regarding the clarity of the reported computation times. We have updated Appendix B.3 to provide the precise specifications and measurement protocols used.
>
> * **Hardware Specifications:** All evaluations were conducted on a standardized setup: an NVIDIA A6000 GPU paired with an Intel Xeon CPU @ 2.20GHz.
> * **Measurement Protocol:**
>     * **Neural Methods (RRNCO, POMO, etc.):** Reported times represent the wall-clock inference time on the GPU with a batch size of 32.
>     * **Traditional Solvers (OR-Tools, LKH3):** Reported times represent CPU processing time, executed sequentially (single thread per instance) to reflect standard benchmarking practices.
> * **Clarification of "7h":** The mention of "7 hours" for OR-Tools refers to the cumulative wall-clock time required to process the entire test set (1,280 instances) sequentially with the configured time limits (e.g., 20 seconds per instance for VRP). In contrast, NCO methods process the same dataset in minutes due to parallel GPU inference. We have clarified this distinction in the text to avoid confusion.
>
> **4. Scope of Generalization and Adaptability to New Benchmarks**
>
> We thank the reviewer for highlighting the critical aspect of generalization to more complex, dynamic settings. We agree that handling time-dependent conditions is a vital "real-world" capability. To demonstrate the extensibility of our approach, we have applied RRNCO to the Stochastic Multi-period Time-dependent VRP (SMTVRP) benchmark (SVRPBench)[1].
>
> * **Multi-Snapshot NAB:** We extended our architecture with a "Multi-Snapshot" Neural Adaptive Bias. Instead of a single duration matrix, this variant processes $K$ distinct duration snapshots (e.g., morning, noon, evening). A learnable gating network fuses these snapshots, allowing the model to "look ahead" and anticipate traffic congestion.
> * **Dynamics-Informed Simulation:** We utilized the SVRPBench protocol, which models congestion using Gaussian kernels and injects stochastic noise (LogNormal) and random incidents (Poisson process).
> * **Results:** As shown in the table below (and added to Appendix E), RRNCO outperforms both traditional and learning-based baselines. Notably, RRNCO maintains 100% feasibility with zero time window violations, whereas OR-Tools fails to find feasible solutions for the majority of instances despite a 1000-second time limit.
>
> | Method | Cost | Feasibility | Runtime | TW Violations |
> | :--- | :---: | :---: | :---: | :---: |
> | ACO (Ant Colony Opt.) | 763.52 | 1.00 | 41.3s | 0.00 |
> | OR-Tools | 610.08 | 0.24 | 1000s | 45.15 |
> | NN + 2-opt | 969.96 | 1.00 | 10s | 0.00 |
> | RouteFinder | 602.20 | 1.00 | 0.05s | 0.00 |
> | GOAL | 1319.00 | 1.00 | 0.28s | 0.00 |
> | **RRNCO (NAB)** | **600.30** | **1.00** | **0.20s** | **0.00** |
> | **RRNCO (Multi-Snapshot)**| **594.19** | **1.00** | **0.20s** | **0.00** |
>
> These results demonstrate that RRNCO is not limited to static asymmetric matrices but can be effectively adapted to dynamic, stochastic, and time-dependent environments, further validating the "real-world" applicability of our proposed mechanisms.
>
> **References:**
> [1] Ahmed Heakl, et al. "SVRPBench: A Realistic Benchmark for Stochastic Vehicle Routing Problem." Thirty-ninth Annual Conference on Neural Information Processing Systems Datasets and Benchmarks Track. (2025)

---

### Official Review · Reviewer_yCJJ · 2025-10-31

**Soundness:** 3
**Presentation:** 3
**Contribution:** 3
**Rating:** 6
**Confidence:** 4

**Summary:**

This paper addresses the critical sim-to-real gap for Neural Combinatorial Optimization (NCO) in Vehicle Routing Problems (VRPs), which stems from oversimplified symmetric data and inadequate node-centric architectures. The authors make a dual contribution: 1)  introducing an Adaptive Node Embedding (ANE) and a Neural Adaptive Bias (NAB) to jointly model real-world asymmetric distance, duration, and angles, and 2) a new, open-source VRP benchmark with asymmetric data from 100 real-world cities. Experiments show RRNCO achieves state-of-the-art performance among NCO solvers on this more realistic benchmark.

**Strengths:**

1. This paper provides an open-source benchmark dataset. This is a major service to the NCO community, enabling the training and validation of NCO solvers on tasks that are more representative of practical settings.

2. The Adaptive Node Embedding (ANE) and Neural Adaptive Bias (NAB)  are effective for fusing multiple asymmetric edge features.

3. The authors compare against a wide range of strong baselines (both traditional and NCO). The evaluation on in-distribution, OOD-city, and OOD-cluster scenarios provides robust support for the authors' claims.

**Weaknesses:**

1. The NAB is a core contribution that models distance, duration, and an "angle matrix".However, the main paper does not define how this angle matrix is computed.

2. The paper fails to specify any of the runtime parameters, configurations, or computational budgets (e.g., time limits) used for these traditional solvers.

3. For dynamic features $D^t$ (defined in Appendix A), it never specifies what these features are for each problem type (ATSP, ACVRP, ACVRPTW). This is a critical missing detail for implementation.

4. In Decoder, the formulation in Line 710 only uses the last node's embedding, this is a significant deviation from standard NCO practice for problems like ATSP (which also use the first node's embedding for context). The paper provides no justification for this non-standard design, making it unclear if it's an intentional choice or an omission.

5. The study does not include a granular ablation on the inputs to NAB. Consequently, it is impossible to determine if the performance gain comes from the novel combination of all three features (D+T+$\Phi$), or if it is overwhelmingly driven by just the distance matrix (D) alone.

6. RRNCO's encoder is based on the AAFM from [1]  and its decoder on [2]. However, these two highly relevant works are not included as baselines in the comparison.

7. The overall quality of this paper is good, but its clarity would benefit from a final proofreading pass to correct some minor typos and inconsistencies:
- In Line 260,  "adaptation" is spelled as "adaption" ;
- The attention layers are referred to as "AAFM" in the text but "AFT Layers" in Table 4;
- The "clipping hyperparameter $C$" from Equation 20 is not defined in the hyperparameter list (Table 4);
- Equation 10 is defined twice (lines 265 and 295);

**Questions:**

1. How is the "angle matrix" $\Phi$, a key input to the NAB (Section 4.1.2), defined and computed?

2. For the classical solver runtimes (LKH3, PyVRP) in Table 1, what was the CPU configuration? Did they utilize multi-core parallel processing, and if so, how many cores were used?

3. Figure 2 is ambiguous. Please clarify the encoder architecture: (1) Is my understanding correct that ANE is a one-time initial embedding step, not part of the attention stack? If so, I suggest revising Figure 2, Figure 2 ambiguously depicts ANE as part of the encoder stack and also omits the multi-layer stacking logic. (2) Why does the encoder diagram omit the Feed-Forward Network (FFN), while the decoder includes one? Does the encoder not require an FFN?

4. In Figure 4, when the NAB is removed, does the model revert to the original heuristic bias, or does it use a different attention mechanism entirely, such as a standard Multi-Head Attention (MHA) block?

References:

[1] Instance-conditioned adaptation for large-scale generalization of neural combinatorial optimization. arXiv preprint arXiv:2405.01906, 2024.

[2] Rethinking Light Decoder-based Solvers for Vehicle Routing Problems. ICLR, 2025.

---

> ### Author Response · Authors · 2025-12-03
> **Official Response to Reviewer yCJJ (Part1)**
>
> We sincerely thank the reviewer for their thoughtful and constructive feedback. We are encouraged by your recognition of our work's contributions, specifically the value of our open source benchmark dataset as a service to the NCO community the effectiveness of our Adaptive Node Embedding (ANE) and Neural Adaptive Bias (NAB) in fusing asymmetric features and the robustness of our evaluation across diverse distribution scenarios.
>
> We have carefully addressed each of your questions and concerns below.
>
> ---
>
> **1. Definition and Computation of the Angle Matrix (Response to W1 & Q1)**
>
> Thank you for pointing out the need for a formal definition of the "angle matrix" in the main text.
>
> To capture the directional relationships between node pairs, which are crucial for understanding spatial structure beyond simple proximity, we introduce the angle matrix $\Phi \in \mathbb{R}^{N \times N}$. The entry $\phi_{ij}$ represents the relative angle from node $i$ to node $j$, computed as:
>
> $$
> \phi_{ij} = \arctan2(y_j - y_i, x_j - x_i)
> $$
>
> where the resulting values lie in the range $(-\pi, \pi]$.
>
> **Intuition:** Relying solely on the distance matrix allows the model to perceive proximity (magnitude) but fails to reconstruct the global spatial structure, as distance is rotation invariant. By incorporating $\Phi$, our model effectively utilizes a polar coordinate system (combining distance magnitude and angular direction). This enables the Neural Adaptive Bias (NAB) to resolve the relative spatial positions of all neighbors globally, effectively acting as a "compass" that guides the attention mechanism through asymmetric landscapes.
>
> We have updated Section 4.1.2 to include this explicit definition and intuition.
>
> **2. CPU Configuration and Parallelization for Classical Solvers (Response to W2 & Q2)**
>
> We appreciate the request for clarification regarding the hardware and runtime settings to ensure fair comparison.
>
> **Hardware Configuration:** All experiments, including classical baselines, were conducted on a server equipped with an NVIDIA A6000 GPU and an Intel Xeon CPU @ 2.20GHz.
>
> **Solver Configuration:**
> To maintain a fair comparison with NCO methods (which are typically evaluated sequentially per batch) and following standard protocols in the literature:
> * **LKH3:** Executed as a single threaded process on a single CPU core with a time limit of 5 seconds per instance.
> * **PyVRP:** Executed as a single threaded process on a single CPU core with a time limit of 20 seconds per instance.
> * **OR-Tools:** Executed as a single threaded process on a single CPU core with a time limit of 20 seconds per instance.
>
> The times reported in Table 1 are the average wall clock times per instance (averaged over the batch size of 32). We have added a detailed hardware and runtime configuration table to **Appendix C** to ensure full reproducibility.
>
> **3. Specification of Dynamic Features (Response to W3)**
>
> Thank you for noting the omission regarding the specific dynamic features used in the decoder context. These features are concatenated with the embedding of the last visited node to form the context vector $h_c$.
>
> The specific dynamic features for each problem type are:
> * **ATSP** No additional dynamic features.
> * **ACVRP:** **[Current Load]**. The remaining capacity of the vehicle is crucial for determining feasibility.
> * **ACVRPTW:** **[Current Load, Current Time]**. Both capacity and the current accumulated time are necessary to satisfy demand and time window constraints.
>
> We have revised Appendix A to explicitly list these features for each problem type to facilitate implementation.
>
> **4. First Node Embedding in ATSP Decoding (Response to W4)**
>
> We fully agree with the reviewer that incorporating the first node into the decoding context is a standard and critical feature for TSP like problems, as the tour must return to the start.
>
> We clarify that **our implementation does indeed include the first node embedding in the context**, consistent with standard NCO practices. The omission of this detail in the manuscript's mathematical formulation was an oversight. We have revised the text in Appendix A to accurately reflect our implementation: the context vector includes the embedding of the first node alongside the last visited node.

---

> ### Author Response · Authors · 2025-12-03
> **Official Response to Reviewer yCJJ (Part2)**
>
> **5. Granular Ablation on NAB Inputs (Response to W5)**
>
> This is an excellent suggestion. To address your concern that performance might be driven solely by the distance matrix, we conducted a granular ablation study on the ACVRPTW task. We progressively removed the duration and angle components from the NAB to isolate their individual contributions.
>
> **Table: Granular ablation on NAB inputs for ACVRPTW**
>
> | Model | In distribution | | OOD (City) | | OOD (Cluster) | |
> | :--- | :---: | :---: | :---: | :---: | :---: | :---: |
> | | **Cost** | **Gap (%)** | **Cost** | **Gap (%)** | **Cost** | **Gap (%)** |
> | **RRNCO Full ($D+T+\Phi$)** | **122.467** | **3.74** | **123.004** | **3.79** | **40.939** | **4.30** |
> | w/o Duration ($D+\Phi$) | 122.693 | 3.93 | 123.249 | 4.00 | 41.077 | 4.65 |
> | w/o Duration & Angle ($D$ only) | 122.849 | 4.06 | 123.364 | 4.09 | 41.151 | 4.84 |
>
> **Analysis:**
> The results demonstrate that while the distance matrix ($D$) provides a strong foundation, the inclusion of Duration ($T$) and Angle ($\Phi$) yields consistent improvements\.
> * Removing **Duration** increases the gap by approximately 0.2% in distribution and 0.35% in OOD Cluster settings.
> * Further removing **Angle** (relying on $D$ only) degrades performance again, increasing the gap to 4.06% (in distribution) and 4.84% (OOD Cluster).
>
> This confirms that the specific combination of all three modalities in the NAB is essential for achieving state of the art performance, particularly in challenging OOD scenarios where the correlation between distance and duration may vary. We have included this new analysis in the revised paper.
>
> **6. Comparison with AAFM and ReLD (Response to W6)**
>
> We appreciate the suggestion to include AAFM and ReLD as direct baselines, given their architectural relevance. We have added these methods to our main comparison table.
>
> **Summary of New Results (ATSP, ACVRP, ACVRPTW):**
>
>
> | Task | Method | In-distribution | | | Out-of-distribution (city) | | | Out-of-distribution (cluster) | | |
> | :--- | :--- | :---: | :---: | :---: | :---: | :---: | :---: | :---: | :---: | :---: |
> | | | **Cost** | **Gap (%)** | **Time (s)** | **Cost** | **Gap (%)** | **Time (s)** | **Cost** | **Gap (%)** | **Time (s)** |
> | **ATSP** | AAFM | 45.992 | 19.812 | 151 | 46.588 | 19.755 | 151 | 15.211 | 24.987 | 151 |
> | | **RRNCO** | **39.077** | **1.797** | **22** | **39.783** | **2.262** | **22** | **12.450** | **2.301** | **22** |
> | | | | | | | | | | | |
> | **ACVRP** | ReLD-MTL | 88.331 | 26.659 | 16 | 88.037 | 24.896 | 16 | 36.169 | 60.373 | 16 |
> | | ReLD-MoEL | 88.154 | 26.406 | 16 | 87.764 | 24.509 | 16 | 36.137 | 60.231 | 16 |
> | | AAFM | 76.663 | 9.928 | 11 | 77.811 | 10.389 | 11 | 25.131 | 11.431 | 11 |
> | | **RRNCO** | **72.145** | **3.450** | **25** | **72.999** | **3.562** | **25** | **23.280** | **3.224** | **25** |
> | | | | | | | | | | | |
> | **ACVRPTW** | ReLD-MTL | 132.722 | 12.423 | 18 | 132.856 | 12.102 | 18 | 51.680 | 31.659 | 18 |
> | | ReLD-MoEL | 132.594 | 12.314 | 18 | 132.621 | 11.904 | 18 | 51.647 | 31.575 | 18 |
> | | **RRNCO** | **122.693** | **3.928** | **35** | **123.249** | **3.996** | **35** | **41.077** | **4.647** | **35** |
>
>
>
> As shown, RRNCO significantly outperforms both the adapted AAFM and ReLD baselines. This highlights that simply applying these architectures to asymmetric data is insufficient; our proposed ANE and NAB mechanisms are critical for effectively processing the complex, multi modal edge features of real world routing. We have updated Table 1 and the Experimental section to include these comparisons.
>
> **7. Typos and Proofreading (Response to W7)**
>
> We thank the reviewer for highlighting the need for a final polish. We have conducted a thorough proofreading of the manuscript to correct typos, improve phrasing, and ensure consistency throughout the text.
>
> **8. Clarification of Figure 2 and Encoder Architecture (Response to Q3)**
>
> We apologize for the ambiguity in Figure 2.
> 1.  **ANE Position:** You are correct. The Adaptive Node Embedding (ANE) is a **one time initial projection step** that generates the input embeddings. It is *not* repeated inside the attention stack.
> 2.  **Encoder FFN:** The encoder layers absolutely include a Feed Forward Network (FFN) following the AAFM block, consistent with standard Transformer style architectures. This is reflected in our training configuration where the feedforward dimension is set to 512.
>
> We have revised Figure 2 to clearly separate the initial embedding block (ANE) from the $N$ layer encoder stack. We have also explicitly visualized the FFN within the encoder layer diagram to remove any confusion.

---

> ### Author Response · Authors · 2025-12-03
> **Official Response to Reviewer yCJJ (Part3)**
>
> **9. NAB Ablation Mechanism (Response to Q4)**
>
> In our ablation study (Figure 4), removing the NAB causes the model to revert to the fixed heuristic bias proposed in the original AAFM paper, defined as $-\alpha \cdot \log(N) \cdot d_{ij}$.
>
> This design choice was intentional to isolate the specific benefit of *learning* the adaptive bias via our NAB module versus using a static, distance based heuristic. The significant performance drop observed when removing NAB confirms that a data driven, multi modal bias is superior to simple distance based heuristics for real world routing.
>
> **References:**
> [1] Fei Liu et al. Multi-Task Learning for Routing Problem with Cross-Problem Zero-Shot Generalization, KDD '24: Proceedings of the 30th ACM SIGKDD Conference on Knowledge Discovery and Data Mining

---

### Official Review · Reviewer_jJZH · 2025-10-31

**Soundness:** 2
**Presentation:** 2
**Contribution:** 2
**Rating:** 4
**Confidence:** 3

**Summary:**

This paper proposes RRNCO, a neural combinatorial optimization architecture aimed to handle the asymmetry in vehicle routing problems. The contributions are twofold: (1) the Adaptive Node Embedding (ANE), which integrates spatial coordinates and asymmetric distance information through a contextual gating mechanism, and (2) the Neural Adaptive Bias (NAB), which jointly models distance, duration, and directional angle features. The authors also construct a new benchmark dataset with asymmetric real-world routing data from OpenStreetMap. Experiments on ATSP, ACVRP, and ACVRPTW demonstrate that RRNCO outperforms prior neural and classical solvers.

**Strengths:**

1. Clear motivation and relevance: the paper addresses a significant practical limitation of current NCO research: the reliance on symmetric, synthetic datasets. The focus on the sim-to-real gap is well-motivated and meaningful for practical deployment.
2. Valuable benchmark contribution: the creation of an asymmetric VRP dataset derived from 100 OpenStreetMap cities represents a substantial and reusable resource for future research. It enhances reproducibility and encourages broader progress in realistic NCO studies.

**Weaknesses:**

1. Limited theoretical novelty: both ANE and NAB are constructed from existing attention and gating mechanisms. While effective, they represent architectural refinements rather than fundamentally new theoretical concepts.
2. Insufficient evaluation for real-world robustness: the sim-to-real gap also arises from dynamic traffic conditions that alter travel times in real operations. The proposed model does not evaluate robustness under such time-dependent or stochastic variations. When congestion occurs after route generation, the model would likely suffer from the same limitations as prior works.

**Questions:**

1. Are there any theoretical analysis about of RRNCO (e.g., learnability, convergence, or generalization bound)?
2. Is it possible to evaluate the robustness of RRNCO with respect to the time-dependent or stochastic variations of traffic conditions?

---

> ### Author Response · Authors · 2025-12-03
> **Official Response to Reviewer jJZH (Part1)**
>
> We sincerely thank the reviewer for their insightful comments and for recognizing the importance of real world routing complexities. We value the suggestion to evaluate time dependent and stochastic variations, which has led us to conduct significant additional experiments that strengthen our submission. Below, we address your concerns regarding theoretical proofs and dynamic evaluation.
>
> ---
>
> **1. Response to W1 and Q1: Theoretical Convergence and Generalization**
>
> We acknowledge that we do not provide formal theoretical proofs for convergence or generalization bounds. However, we clarify that the primary contribution of RRNCO is architectural and empirical, focusing on designing effective inductive biases for real world routing complexities where classical theoretical bounds often do not apply. This approach aligns with the established standards of the Neural Combinatorial Optimization (NCO) literature, such as the Attention Model, POMO, and MatNet, where advancements are driven by architectural innovations validated through rigorous empirical evaluation.
>
> **Architectural Inductive Bias as Justification**
>
> While we lack formal proofs, our design choices are grounded in the specific multi modal nature of real world VRPs, which we argue provides a strong constructive justification for the model's performance:
>
> * **Adaptive Node Embedding (ANE):** Real world routing often presents conflicting signals between coordinate spaces (Euclidean geometry) and road network graph spaces (actual travel cost). ANE employs a learned contextual gating mechanism to fuse these distinct modalities dynamically, rather than relying on simple concatenation.
>
> * **Neural Adaptive Bias (NAB):** A critical limitation of previous node based architectures is their inability to efficiently integrate complex edge features. RRNCO is the first architecture to jointly model relative directional angles, asymmetric distance, and the duration matrix via NAB.
>     * *Why this matters:* In real world logistics (specifically VRPTW), the solver must optimize for one metric (minimizing total fleet travel distance) while strictly adhering to constraints governed by another metric (travel duration for time windows).
>     * By fusing these distinct modalities into the attention mechanism via the learned bias matrix $A$, NAB enables the model to weigh spatial efficiency against temporal feasibility dynamically. This allows the solver to make routing decisions that minimize distance costs without violating time window constraints, a capability that standard Euclidean based or single cost matrix models struggle to achieve.
>
> **Empirical Generalization**
>
> We demonstrate generalization empirically, which serves as practical evidence of the model's robustness. As shown in Table 1 and Figure 4 of the main paper, RRNCO generalizes effectively to:
> 1.  **Out of Distribution (OOD) Cities:** Maps not seen during training.
> 2.  **OOD Clusters:** Different node distributions.
> 3.  **Cross-Task:** Solving ATSP, ACVRP, and ACVRPTW with a unified architecture.

---

> ### Author Response · Authors · 2025-12-03
> **Official Response to Reviewer jJZH (Part2)**
>
> **2. Response to W2 and Q2: Evaluation under Time Dependent or Stochastic Traffic Variations**
>
> We thank the reviewer for highlighting this critical aspect of real world deployment. We agree that handling dynamic conditions, such as rush hour congestion and stochastic delays, is essential. We are pleased to report that RRNCO's architecture is natively designed to support these complexities, and we have validated this through additional experiments using the SVRPBench[1] protocols.
>
> **Architectural Extension: Multi Snapshot Neural Adaptive Bias (NAB)**
>
> To demonstrate the flexibility of our framework, we extended the core Neural Adaptive Bias (NAB) to a **Multi Snapshot NAB**. While the standard NAB fuses a single duration matrix, this extension processes $K$ distinct "duration snapshots" $\{T_1, \dots, T_K\}$ representing predicted traffic conditions (e.g., morning peak, noon, evening rush).
> * *Mechanism:* As detailed in Appendix D.4, we employ a learnable gating network that dynamically computes weights for distance, angle, and each duration snapshot. This allows the model to **look ahead** and effectively fuse the traffic layers most relevant to the **global routing context**, enabling the decoder to handle time-dependent variations implicit in the encoded representation.
>
> **Dynamics-Informed Traffic Simulation (SVRPBench)**
>
> We implemented a Dynamics-Informed Traffic Simulator strictly adhering to the Time Dependent Travel Time Modeling proposed in SVRPBench. The simulator models time dependent congestion using Gaussian kernels and injects stochastic noise (LogNormal distribution) and random accident events (Poisson processes) to simulate real world uncertainty.
>
> **Experimental Results**
>
> We evaluated our model on VRPTW instances ($N=50$) augmented with time dependent durations. The results (referencing the baseline protocols in Appendix D) are presented below:
>
> | Method | Cost | Feasibility | Runtime | TW Violations |
> | :--- | :---: | :---: | :---: | :---: |
> | ACO (Ant Colony Opt.) | 763.52 | 1.00 | 41.3s | 0.00 |
> | OR Tools | 610.08 | 0.242 | 1000s | 45.15 |
> | NN + 2opts | 969.96 | 1.00 | 10s | 0.00 |
> | RouteFinder | 602.20 | 1.00 | 0.05s | 0.00 |
> | GOAL | 1319.00 | 1.00 | 0.28s | 0.00 |
> | **RRNCO (NAB)** | **600.30** | **1.00** | **0.20s** | **0.00** |
> | **RRNCO (Multi Snapshot NAB)** | **594.19** | **1.00** | **0.20s** | **0.00** |
>
> **Analysis:**
>
> * **Superior Performance:** The standard RRNCO (NAB) is already highly competitive. However, the Multi Snapshot NAB achieves the lowest cost (594.19), validating that our encoder can successfully capture high dimensional correlations in dynamic environments.
> * **Feasibility vs. Baselines:** Notably, traditional solvers like OR Tools struggled significantly with the hard time constraints of the dynamic setting (resulting in high violations), whereas RRNCO maintains 100% feasibility with very low inference time.
>
> We have included these full experimental details, simulation methodology, and results in Appendix E of the revised paper.
>
>
> **References:**
> [1] Ahmed Heakl, et al. "SVRPBench: A Realistic Benchmark for Stochastic Vehicle Routing Problem." Thirty-ninth Annual Conference on Neural Information Processing Systems Datasets and Benchmarks Track. (2025)

---

### Author Response · Authors · 2025-12-03
**General Responses and Summary of Revisions**

Dear Reviewers/AC/SAC/PC

We thank you and all reviewers for the thoughtful and constructive feedback. We are glad that reviewers highlighted several strengths of our work, including:
- **Clear motivation and relevance** of addressing the sim-to-real gap in NCO (Reviewer jJZH, Reviewer yCJJ, Reviewer mJYV),
- **High community value** of the new large-scale real-world benchmark (Reviewer jJZH,, Reviewer yCJJ, Reviewer mJYV, Reviewer XMP3),
- **Effectiveness of ANE and NAB** for modeling asymmetric, multi-modal routing features (Reviewer yCJJ, Reviewer XMP3), and
- **Strong empirical results and robust generalization** across in-distribution and OOD scenarios (Reviewer yCJJ).

Below we summarize how we addressed the primary concerns:

1. **Dynamic and stochastic environments (Reviewer jJZH, Reviewer mJYV):**
   We conducted significant new experiments using the **SVRPBench** protocol and introduced a **Multi-Snapshot NAB** variant that handles time-dependent traffic. RRNCO maintains 100% feasibility and outperforms classical solvers under dynamic conditions. Full results are now included in the appendix E.

2. **Missing definitions and architectural clarity (Reviewer yCJJ):**
   We added precise definitions for the angle matrix, specified the dynamic features for each task, clarified that the decoder *does* use the first-node embedding, and revised Figure 2 to cleanly separate ANE from the encoder and to show the FFN modules.

3. **Benchmarking completeness (Reviewer yCJJ, Reviewer mJYV)):**
   We added new comparisons including **OR-Tools for ATSP**, **AAFM**, and **ReLD**, showing RRNCO consistently achieves lower cost and far better OOD robustness while being significantly faster.

4. **Ablations on NAB inputs (Reviewer yCJJ):**
   We added a granular ablation demonstrating that duration and angle features each provide measurable gains, confirming the necessity of NAB’s multi-modal bias rather than distance alone.

5. **Clarity and reproducibility (Reviewer yCJJ, Reviewer yCJJ):**
   We clarified all classical solver runtimes, CPU/GPU configurations, and time-limit settings, and performed a thorough proofreading to resolve typos and minor inconsistencies.

We believe these additions substantially strengthen the submission, address all major reviewer concerns, and further demonstrate RRNCO's practical value and extensibility to real-world routing settings.

We thank you again for your time and consideration.


Best regards,

RRNCO's Authors

---

### Meta-Review · Area_Chair_H9AK · 2026-01-01

**Summary:**

This paper proposes a neural combinatorial optimization architecture named RRNCO to handle the asymmetry in vehicle routing problems. The two introduced comments are the Adaptive Node Embedding (ANE) and the Neural Adaptive Bias (NAB). A new benchmark dataset with asymmetric real-world routing data is also constructed from OpenStreetMap. Experiments on ATSP, ACVRP, and ACVRPTW show that RRNCO outperforms prior neural and classical solvers.

All reviewers acknowledged the value of this work, but also raised some major concerns, e.g., insufficient evaluation for real-world robustness, missing comparison to some baselines, and generalization to other vrp benchmarks. The authors gave detailed responses, and added additional experimental results. I think the concerns have been addressed to some extent.

**Reviewer Concerns:**

Reviewer jJZH has major concerns: 1) limited novelty (as both ANE and NAB are constructed from existing attention and gating mechanisms). 2) Insufficient evaluation for real-world robustness.

The authors gave detailed responses, and added additional experimental results. I think some of the concerns (e.g., insufficient evaluation for real-world robustness) have been addressed.

Reviewer yCJJ mainly has major concerns: 1) missing details for some implementations; 2) more ablations and baselines need to be included in the empirical comparison.

The authors gave detailed responses, and added additional experimental results. I think the concerns have been addressed.

Reviewer mJYV has major concners: 1) uncomprehensive problem definition; 2) missing comparison to or-tools in atsp; 3) unclear computation time; 4) generalization to other vrp benchmarks.

The authors gave detailed responses, made clarifications, and added additional experimental results. I think the concerns have been addressed to some extent.

Reviewer XMP3 gave the highest score, and only raised the main question: Can the adaptive attention-free module handle features other than travel time and distances?

The authors gave positive answers, and provided explanations.

**Reviewer Scores:**

Reviewer jJZH may increase the score as the authors provided additional experiments for real-world robustness.

Reviewer yCJJ may keep the positive score, as the authors gave detailed responses and added additional experimental results.

Reviewer mJYV may increase the score, as the authors gave detailed responses, made clarifications, and added additional experimental results. I think most of the concerns can be addressed to some extent.

Reviewer XMP3 may keep the positive score, as the authors answered her/his question well.

---

### Decision · Program_Chairs · 2026-01-26

Accept (Poster)